# Integration of light and metabolic signals for stem cell activation at the shoot apical meristem

Anne Pfeiffer[1], Denis Janocha[1], Yihan Dong[2], Anna Medzihradszky[1], Stefanie Schöne[1], Gabor Daum[1], Takuya Suzaki[1†], Joachim Forner[1], Tobias Langenecker[3], Eugen Rempel[1], Markus Schmid[3‡], Markus Wirtz[2], Rüdiger Hell[2], Jan U Lohmann[1]*

[1]Department of Stem Cell Biology, Centre for Organismal Studies, Heidelberg University, Heidelberg, Germany; [2]Department of Molecular Plant Biology, Centre for Organismal Studies, Heidelberg University, Heidelberg, Germany; [3]Department of Molecular Biology, Max Planck Institute for Developmental Biology, Tübingen, Germany

**Abstract** A major feature of embryogenesis is the specification of stem cell systems, but in contrast to the situation in most animals, plant stem cells remain quiescent until the postembryonic phase of development. Here, we dissect how light and metabolic signals are integrated to overcome stem cell dormancy at the shoot apical meristem. We show on the one hand that light is able to activate expression of the stem cell inducer *WUSCHEL* independently of photosynthesis and that this likely involves inter-regional cytokinin signaling. Metabolic signals, on the other hand, are transduced to the meristem through activation of the TARGET OF RAPAMYCIN (TOR) kinase. Surprisingly, TOR is also required for light signal dependent stem cell activation. Thus, the TOR kinase acts as a central integrator of light and metabolic signals and a key regulator of stem cell activation at the shoot apex.

*For correspondence: jan. lohmann@cos.uni-heidelberg.de

Present address: †Graduate School of Life and Enviromental sciences, University of Tsukuba, Tsukuba, Japan; ‡Umeå Plant Science Centre, Umeå University, Umeå, Sweden

**Competing interests:** The authors declare that no competing interests exist.

## Introduction

Light is the sole energy source of plants and therefore one of the most important environmental factors influencing their development and physiology. Consequently, several of the core developmental decisions during the lifecycle of a plant from germination to seedling development and flowering are strongly influenced by light conditions. After germination, higher plants undergo two distinct developmental programs depending on the availability of light, termed skotomorphogenesis and photomorphogenesis. Skotomorphogenesis, the dark adaptation program, is characterized by an etiolated phenotype, including an elongated hypocotyl, closed cotyledons, the formation of an apical hook and etioplast development. Importantly, stem cells at the shoot and root tip remain dormant and thus growth in etiolated seedlings is mainly dependent on cell elongation rather than cell division. In contrast, photomorphogenesis, the developmental program triggered in light, leads to seedlings with short hypocotyls, unfolded cotyledons and development of chloroplasts. In the light, shoot and root meristems are activated, leading to root growth and development of the first leaves by cell division and expansion (reviewed in *Nemhauser and Chory 2002*).

Based on evolutionary evidence, photomorphogenesis is the default pathway, since gymnosperms for example do not follow a strict skotomorphogenic development in darkness (*Wei, 1994*). With the advance of land plants and resulting new environmental challenges, such as growth in dense canopy and germination in soil, the evolution of the dark-adapted skotomorphogenesis

**eLife digest** Plants are able to grow and develop throughout their lives thanks to groups of stem cells at the tips of their shoots and roots, which can constantly divide to produce new cells. Energy captured from sunlight during a process called photosynthesis is the main source of energy for most plants. Therefore, the amount and quality of light in the environment has a big influence on how plants grow and develop. An enzyme called TOR kinase can sense energy levels in animal cells and regulate many processes including growth and cell division. Plants also have a TOR kinase, but it is less clear if it plays the same role in plants, and whether it can respond to light.

Plant stem cells only start to divide after the seed germinates. In shoots, a protein called WUSCHEL is required to maintain stem cells in an active state. Here, Pfeiffer et al. studied how shoot stem cells are activated in response to environmental signals in a plant known as *Arabidopsis*. The experiments show that light is able to activate the production of WUSCHEL independently of photosynthesis via a signal pathway that depends on TOR kinase. The stem cells do not directly sense light; instead other cells detect the light and relay the information to the stem cells with the help of a hormone called cytokinin.

Further experiments show that information about energy levels in cells is relayed via another signal pathway that also involves the TOR kinase. Therefore, Pfeiffer et al.'s findings suggest that the activation of TOR by light allows plant cells to anticipate how much energy will be available and efficiently tune their growth and development to cope with the environmental conditions. Future challenges are to understand how TOR kinase is regulated by light signals and how this enzyme is able to act on WUSCHEL to trigger stem cell division.

program ensued an advantage: It allowed plants to allocate the limited energy sources supplied by the seed to maximally grow by elongation, in order to reach favorable light conditions that will provide energy for further growth and development. To faithfully execute these opposing developmental programs, plants have evolved complex mechanisms to perceive light quality and quantity through a whole range of photoreceptors that are mainly absorbing in the blue, red and far-red range of the spectrum. Activation of the blue absorbing CRYPTOCHROMES (crys) and/or the red and far-red absorbing PHYTOCHROMES (phys) overrides the skotomorphogenic program and plants undergo photomorphogenesis within minutes after perception of a light stimulus (reviewed in *Chory, 2010*). On the molecular level, activated light receptors inhibit the function of the core repressor of photomorphogenesis, CONSTITUTIVE PHOTOMORPHOGENESIS 1 (COP1), an E3 ubiquitin ligase that targets positive regulators of photomorphogenesis for degradation in darkness (*Yi and Deng, 2005*). At the same time, a group of potent transcription factors, the PHYTO-CHROME INTERACTING FACTORS (PIFs), which promote skotomorphogenesis in darkness, are degraded upon light perception through the PHYTOCHROMES (*Leivar and Quail, 2010*). The activities of these pathways converge on the differential regulation of thousands of genes resulting in a massive reprogramming of the transcriptome in response to light (*Ma et al., 2002*; *Tepperman et al., 2004*; *Peschke and Kretsch, 2011*; *Pfeiffer et al., 2014*).

Light not only activates photoreceptors, it also fuels photosynthesis and therefore leads to the production of a number of energy rich metabolites including sugars. Plants are able to monitor their metabolic state with several signaling systems (*Lastdrager et al., 2014*) and recent studies have focused on the evolutionary conserved TARGET OF RAPAMYCIN (TOR) kinase complex (*Dobrenel et al., 2016*). In other eukaryotes, TOR functions as a central integrator of nutrient, energy, and stress signaling networks and consistently, TOR regulates cell growth and proliferation, ribosome biogenesis, protein synthesis, cell wall integrity and autophagy (*Díaz-Troya et al., 2008*; *Henriques et al., 2014*; *Lastdragere et al., 2014*; *Xiong and Sheen, 2014*). While other eukaryotes possess two TOR complexes, so far only a single complex has been identified in plants. It is comprised of TOR, FKBP12, LST8 and RAPTOR (*Mahfouz et al., 2006*; *Moreau et al., 2012*) and thus resembles the mammalian TOR complex 1 (mTORC1). AtTOR is expressed in the embryo and endosperm and in meristematic regions of the adult plant (*Menand et al., 2002*). While *tor* null mutants show premature arrest of embryo development (*Menand et al., 2002*), knock down of TOR leads to

growth reduction and affects the carbohydrate and amino acid metabolism (*Caldana et al., 2013*). In contrast, the presence of sugars in general promotes TOR kinase activity (*Ren et al., 2012*; *Dobrenel et al., 2013*; *Xiong et al., 2013*). So far, the only known direct downstream targets of AtTOR kinase are S6 kinase 1 (S6K1) (*Schepetilnikov et al., 2011*, *2013*; *Xiong et al., 2013*), TAP46 (*Ahn et al., 2011*, *2014*) and E2 promoter binding factor a (E2Fa) (*Xiong et al., 2013*). S6K1 plays an important role in reinitiating translation (*Schepetilnikov et al., 2011*) as well as in the regulation of the cell cycle (*Henriques et al., 2010*; *Shin et al., 2012*). Similarly, E2Fa is associated with cell cycle control through the expression of S-phase genes (*Polyn et al., 2015*). Though little is known about how TOR is activated on a molecular level in plants, reports from the past decade suggest that TOR functions as a central regulator of protein synthesis, cell proliferation and metabolism in response to metabolic signals.

Several photomorphogenic responses, like the inhibition of hypocotyl elongation, unfolding of the hypocotyl hook and cotyledons, as well as chloroplast development can be triggered by a light signal alone, as displayed in dark-grown *cop1* and *pif1/pif3/pif4/pif5* quadruple mutants (*pifq*) (*Deng and Quail, 1991*; *Leivar et al., 2009*). However, root growth is not induced in *cop1* mutants unless sucrose is supplied with the growth medium. Photosynthetic assimilates dominantly promote growth in the root where they can synergistically interact with photoreceptor-triggered light signaling (*Kircher and Schopfer, 2012*). Recently, Xiong et al. showed that this photosynthesis-driven growth and proliferation in the root is mediated by the TOR kinase (*Xiong et al., 2013*).

Here, we analyzed the role of light and nutrients for post-germination stem cell activation in the shoot apical meristem (SAM) of young seedlings. Stem cell control in the SAM of *Arabidopsis thaliana* is based on the activity of the homeodomain transcription factor *WUSCHEL (WUS)*, which is expressed in the organizing centre and necessary and sufficient to non-cell- autonomously induce stem cell fate by protein movement. Stem cells in turn express CLAVATA3 (CLV3), a short secreted peptide, that acts via the CLV / CORYNE receptor system to limit the expression of WUS in the organizing center (*Schoof et al., 2000*; *Daum et al., 2014*). The use of a reporter system based on the regulatory regions of *WUS* and *CLV3* allowed us to quantitatively trace behavior of stem cells (*pCLV3:mCHERRY-NLS*) as well as cells of the underlying organizing center (*pWUS:3xVENUS-NLS*). With the help of these tools, we were able to genetically dissect the individual contribution of light signaling and photosynthesis-driven nutrient sensing on the stem cell system of the SAM. We show that both pathways ultimately converge at the level of TOR kinase activation, revealing a role for TOR as a central regulator of stem cell activation in response to environmental cues.

## Results

### Organogenic development is dependent on light and energy metabolites

The SAM of *Arabidopsis* seedlings remains dormant during skotomorphogenesis and therefore, plants are unable to advance to the organogenic stage in the absence of light. However, since light acts as signal and energy source alike, we first asked which of the two roles is dominant for SAM development. While supplementation of sugar to wild-type seedlings grown in the dark is known to be inefficient to trigger development (*Figure 1A*), activation of the light pathway alone, either physiologically by low level illumination, or genetically by introduction of the *cop1* mutation, was shown to induce photomorphogenic development of the hypocotyl and cotyledons in darkness (*Deng and Quail, 1991*). Despite this stark developmental transition, SAMs of *cop1* mutants were unable to produce organs when grown in the dark. However, the SAM was activated and organogenesis initiated in 100% of the dark-grown *cop1* mutants when supplemented with sucrose as external energy source (*Figure 1B*, see also *McNellis et al., 1994*; *Nakagawa and Komeda, 2004*). Conversely, the SAM of light-grown wild-type seedlings remained dormant when photosynthesis was compromised by the carotenoid biosynthesis inhibitor norflurazon. In line with our observation of *cop1* mutants, supplementing the growth medium of these plants with sucrose rescued the dormant phenotype in approximately every third seedling (*Figure 1C*). Thus, neither the availability of energy metabolites, nor light perception alone was sufficient for SAM activation. In contrast, light and energy, likely in the form of photosynthetic products, seemed to be sensed independently, and both factors need to act cooperatively to trigger SAM development.

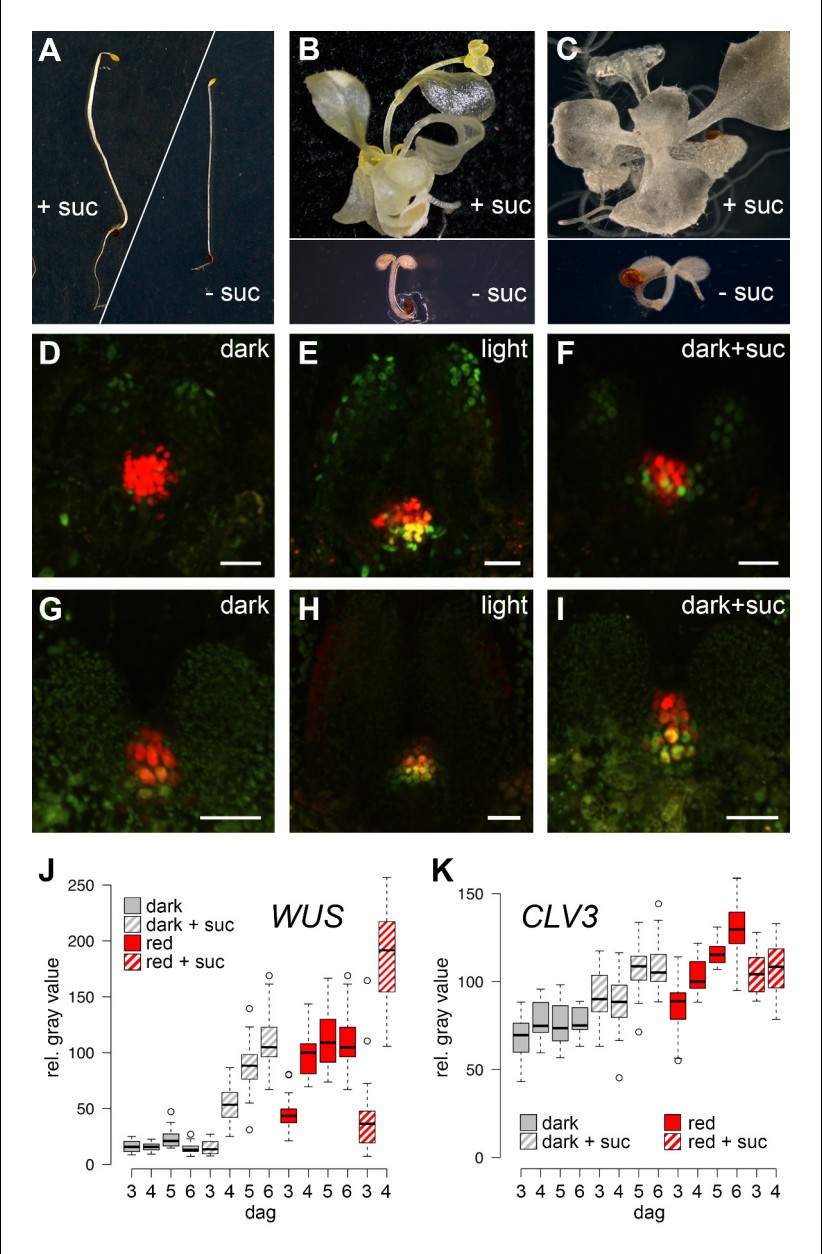

**Figure 1.** SAM development depends on light and sugar. (A–C) Five week old plants grown on media with (+) or without (-) sucrose. (A) Wild-type plants grown in darkness, (B) *cop1* mutant plants grown in darkness and (C) wild-type seedlings grown in light in the presence of 0.5 μM photosynthesis inhibitor norflurazon . (D–I) Maximum projections of SAMs of four day old seedlings; scale bar 20 μm. (D–F) *pCLV3:mCHERRY-NLS* (red) and *pWUS:3xVENUS-NLS* (*green*). (G–I) *pCLV3:mCHERRY-NLS* (red) and *pWUS:WUS-linker-GFP* (green). Quantification of *pWUS:3xVENUS-NLS* (J) and *pCLV3:mCHERRY-NLS* (K) expression by fluorescence intensity under different growth conditions (gray = darkness, red = red light (30 μmol*m$^{-2}$*s$^{-1}$), solid box = w/o sucrose, hatched box = 1% sucrose, dag = days after germination).

The following figure supplement is available for figure 1:

**Figure supplement 1.** Expression of *CLV3* and *WUS* during seedling development.

## WUS expression is independently regulated by light and sucrose

Our phenotypic analysis suggested that light and metabolic signals synergize to activate SAM development, and thus we asked which of the known components underlying stem cell homeostasis might be the relevant cellular and molecular targets. By using transcriptional reporters for stem cells (*pCLV3:mCherry-NLS*) and niche cells (*pWUS:3xVENUS-NLS*) we found that stem cell identity was actively maintained independently of growth conditions and was even observed in the dormant state mediated by germination in the dark (*Figure 1D*). In contrast, expression of the reporter for the stem cell inducing WUS transcription factor was critically dependent on environmental signals and preceded meristem activity and the initiation of organogenic development (*Figure 1E*). To test if our *WUS* reporter faithfully recapitulated the behavior of the endogenous gene, we used in situ hybridization and were able to confirm strong light dependent induction of *WUS* mRNA (*Figure 1—figure supplement 1C and D*). Since WUS protein exhibits complex movement and a short lifetime (*Daum et al., 2014*), we furthermore analyzed the behavior of WUS-GFP protein in vivo by recording the GFP signal in our rescue line (*pWUS:WUS-linker-GFP* in *wus* mutant background [*Daum et al., 2014*]). Again, we observed a strong light- and sucrose-dependency of the WUS-GFP signal in line with the observed activation of the *WUS* promoter and accumulation of the endogenous *WUS* mRNA under these conditions (*Figure 1G–I*) confirming that the simple *pWUS:3xVenus-NLS* reporter represents a faithful and quantitative readout for WUS activity. Taken together, these findings on the one hand suggested that *CLV3* expression is at least partially independent of WUS and on the other hand that the environmentally dependent transcriptional activation of *WUS* is the trigger to overcome stem cell dormancy.

Using seedlings carrying both reporters grown under wave-length specific LED illumination and image quantification we found that the *WUS* reporter (*pWUS:3xVENUS-NLS*) was below detection level in dark-grown seedlings. In contrast, GFP signals were readily detectable in light-grown plants from three days after germination onwards with the signal steadily increasing over time (*Figure 1J*). Interestingly, *WUS* expression was also induced in the absence of light, when plants were grown on sucrose-supplemented medium (*Figure 1F,J*) and when sucrose was supplied to light-grown seedlings, the effect of light and sucrose on *WUS* expression was additive (*Figure 1J*). Light and sucrose also had a similar effect on the regulation of *CLV3* expression (*Figure 1D–I, K*), however, since the *CLV3* reporter was already detectable in dark-grown seedlings, the induction of expression by light and sucrose was less pronounced and mainly due to an enlargement of the *CLV3* domain rather than an increase of signal intensity in individual cells (*Figure 1—figure supplement 1A and B*). In sum, development of the SAM required both, light signal transduction and the availability of photosynthetic products, whereas *WUS* expression was induced also by each signal individually. Thus, tracing *WUS* expression in the SAM of young seedlings represented a sensitive model to decipher the contribution of upstream signals to stem cell activation in a developmentally and physiologically relevant setting.

## Mechanisms of light dependent stem cell activation

Since the expression of the transcriptional *WUS* reporter showed an early and dynamic response to environmental stimuli that mimicked both endogenous *WUS* mRNA, as well as WUS-GFP protein, we used the intensity of the reporter signal in four day-old seedlings as an easily quantifiable proxy for stem cell activation. First, we wanted to elucidate the molecular players involved in stem cell activation by light. To this end, we irradiated seedlings with monochromatic light of low intensities (30 $\mu mol*m^{-2}*s^{-1}$) to analyze the effect of light signaling with minimal influence of photosynthesis-derived metabolites. Even at low intensities, blue, as well as red light were sufficient to robustly induce *WUS* expression (*Figure 2A*). In line with the well-documented biochemistry of the photoreceptors, red-light-induced *WUS* activation was specifically reduced in the *phyB* mutant background, while blue-light-induced reporter activity was impaired in the *cry1/cry2* double mutant background (*Figure 2A*). We thus concluded that light perceived through phyB as well as the crys influences the developmental fate of the SAM.

We also tested *WUS* expression under far-red light, which is sensed by phyA and found that reporter activity was only weakly induced by far-red light. Interestingly, *phyA* mutants showed similar *WUS* promoter activity under far-red light and in darkness (*Figure 2—figure supplement 1A*). However, when we supplemented the growth media with 1% sucrose we observed a clear induction of

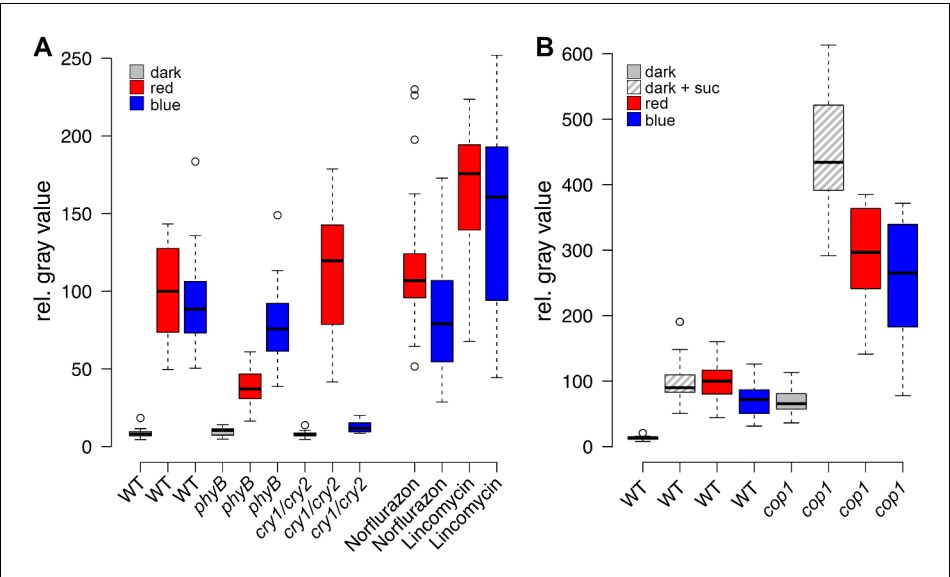

**Figure 2.** Light induced *WUS* expression depends on photoreceptors and is repressed by COP1. Quantification of *pWUS:3xVENUS-NLS* expression by fluorescence intensity was measured in four day old wild-type (WT) or mutant seedlings (WT) or mutant background under different growth conditions (gray = darkness, red = red light (30 μmol*m$^{-2}$*s$^{-1}$), blue = blue light (30 μmol*m$^{-2}$*s$^{-1}$), solid box = w/o sucrose, hatched box = 1% sucrose). (A) 0.5 mM lincomycin and 5 μM norflurazon, respectively were applied to the growth media of wild-type seedlings. (B) Impact of *cop1−4* mutation on WUS expression.

The following figure supplement is available for figure 2:

**Figure supplement 1.** Light regulation of *WUS* expression.

*WUS* expression in response to far-red light, which was dependent on a functional copy of *PHYA*. Still, *phyA* mutants displayed a basal level of *WUS* promoter activity already in darkness even when grown on plates containing sucrose, suggesting a complex and so far unknown regulatory role for phyA under these conditions.

The fact that plants grown in far-red light are photosynthetically inactive and required an exogenous energy source for *WUS* activation, while plants under blue and red light did not, raised the question whether minimal levels of photosynthetically derived sugars might contribute to *WUS* expression in blue and red light, despite the low fluence. Therefore, we tested whether the availability of photosynthetic products is a prerequisite for light-induced *WUS* expression by chemical interference. However, the inhibition of photosynthesis by either norflurazon or lincomycin, did not affect *WUS* promoter activity in red or in blue light (*Figure 2A*). In the presence of lincomycin, *WUS* expression was even slightly increased under both light conditions. To avoid potential side effects of the pharmacological treatments we also tested the effect of $CO_2$ withdrawal on seedling development and *WUS* expression. Preventing photosynthetic assimilation in a $CO_2$-deficient atmosphere inhibited development of seedlings even when grown in light. This phenotype could be rescued in one third of the plants by adding 1% sucrose to the media, similar to what we observed using norflurazon treatment (compare *Figure 1C* and *Figure 2—figure supplement 1C*). Importantly, *WUS* induction by red light was unaffected by $CO_2$ reduction in the atmosphere (*Figure 2—figure supplement 1B*). Thus, photosynthetically derived metabolites produced in a low light environment were not required for activation of stem cells, confirming that light signaling alone was sufficient for *WUS* expression.

We next asked how the light signal perceived by PHYTOCHROMES and CRYPTOCHROMES is relayed to the nucleus by testing the contribution of known downstream signaling components, such as COP1 and HY5. The E3-ubiquitin ligase COP1, which targets HY5 but also other factors for degradation in darkness, showed robust inhibitory effects on *WUS* expression. *Cop1-4* mutants displayed

photomorphogenic development in darkness, which was accompanied by *WUS* expression (*Figure 2B*). Furthermore, the repressive function of COP1 was prominent under all conditions tested and *cop1-4* seedlings displayed strongly elevated *WUS* promoter activity compared to wild-type when grown in dark with or without sugar, and also under low light conditions (*Figure 2B*). To confirm that these effects were not caused by second site mutations present in the *cop1-4* background or specific to the allele tested, we used qRT-PCR to assay *WUS* expression in seedlings carrying other *cop1* loss-of-function alleles. However, since this approach lacked the spatial resolution provided by microscopic quantification of the *WUS* reporter, it proofed to be much less sensitive. Still, we were able to detect accumulation of endogenous *WUS* mRNA in response to light in 7d old wild-type seedlings, as well as in *cop1-4* mutants in the dark (*Figure 2—figure supplement 1D*). Importantly, all three *cop1* mutant alleles tested showed robust elevation of *WUS* mRNA levels when grown in the dark (*Figure 2—figure supplement 1E*), demonstrating that loss of COP1 function leads to activation of WUS.

One of the main functions of COP1 is to target the transcription factor HY5, a positive master regulator of photomorphogenesis, for degradation. Thus, we analyzed the role of HY5 working under the hypothesis that in contrast to *cop1* mutants, which had shown elevated *WUS* reporter expression, *hy5* mutants should suffer from a much reduced meristem activity due to the absence of an important photomorphogenesis stimulating activity. However, *hy5* mutants were unaffected in activation of *WUS* expression (*Figure 2—figure supplement 1F*), suggesting that SAM stem cell activation is dependent on another COP1-targeted transcriptional transducer, such as HY5 HOMO-LOG (HYH), or a so far unknown regulator.

Since the SAM is shielded from the environment especially in etiolated seedlings, where it is buried between the closed cotyledons and protected by the apical hook of the hypocotyl, it seemed questionable that the meristem itself is the site of light perception. We therefore tested the competence of different tissues to perceive light signals and translate them into a stem cell activating output. To this end, we expressed a constitutive active form of phyB (*Su and Lagarias, 2007*) under different tissue specific promoters (*Figure 2—figure supplement 1C,D*). Expression of phyB Y276H under an ubiquitous promoter (*pUBI10*) caused strong *cop1*-like phenotypes and a substantial activation of the *WUS* promoter in the SAM showing that transgenic activation of light signaling is sufficient to trigger stem cell activation in darkness (*Figure 2—figure supplement 1G,H,J*). In line with our hypothesis that light is likely perceived by cells outside the SAM, vascular specific expression of phyB Y276H by the *pSUC2*, or mesophyll specific expression by *pCAB3* promoters (*Ranjan et al., 2011*) initiated constitutive photomorphogenic phenotypes and *WUS* expression in dark-grown seedlings. Similar results were also obtained for the epidermal *pML1* promoter, in lines showing high expression levels of phyB Y276H (*Figure 2—figure supplement 1G–J*). These results suggested that the stimulus downstream of light perception can be transmitted between tissues by a mobile signal and raised the question whether the SAM itself even has the ability to respond to light. To explore this, we expressed phyB Y276H specifically in the SAM under the promoter of *At3g59270* (*Yadav et al., 2009*), but in contrast to expression outside of the SAM, we observed fully etiolated seedlings without detectable *WUS* expression when these plants were grown in the dark. Even when we drove *PHYB Y276H* expression in cells surrounding the organizing center by the promoter of *At1g26680* (*Yadav et al., 2009*), we only observed a minor reduction in hypocotyl elongation in darkness compared to wild-type and marginal *WUS* expression (*Figure 2—figure supplement 1G, H,J*). We therefore concluded that light is perceived by cells outside of the SAM, likely in the cotyledons or the hypocotyl and that this stimulus is transmitted to the SAM by a so far unidentified mobile signal. Amazingly, the SAM does not possess the competence to perceive and/or translate the light stimulus into stem cell activation, but rather is limited to responding to the signals that are transmitted from distant plant organs.

## Mechanisms of hormonal stem cell activation

Since we had shown that light is perceived at a distance from the SAM, which also for energy rich metabolites is not a source, but a sink tissue, we next asked how the information for both environmental inputs is relayed to the stem cell system. Obvious candidates for inter-regional signaling components are plant hormones and there are a number of studies demonstrating their importance in regulating the shoot stem cell niche, especially for cytokinin (CK) and auxin (reviewed in *Murray et al., 2012*). A previous study had analyzed the environmental influence on organ initiation

at the SAM using transfer of light grown tomato plants to darkness as a model and found that light is required for CK signaling and polarized membrane localization of the auxin export carrier PIN1 (*Yoshida et al.,2011*). However, these studies could not distinguish whether light was perceived as informational cue or energy source. We therefore analyzed CK signaling activity using the *pTCSn: GUS* cytokinin output sensor (*Zürcher et al., 2013*) as well as auxin flux directionality using polarization of *pPIN1:PIN1-GFP* (*Benková et al., 2003*) as a proxy (*Figure 3A–K*). In line with the findings of Yoshida et al., CK signaling was strongly activated by light when compared to etiolated seedlings (compare *Figure 3H,G*). Furthermore, we also found that PIN1 polarly localized to the plasma membrane in a light dependent manner (*Figure 3A and B*). Interestingly, sucrose treatment of etiolated seedlings did not affect the localization of PIN1 (*Figure 3C*) but lead to a mild activation of the *TCS* reporter also in the absence of light (*Figure 3I*), suggesting that there is specificity in the hormonal response.

To test the light signaling response independently from impeding effects of photosynthesis, we treated plants grown in light with the photosynthesis inhibitor norflurazon. While PIN1 localization was still light responsive, no activity of the *TCS* reporter was detectable under these conditions (*Figure 3D,J*). We also observed a reduction of *TCS* signal when plants were grown in a $CO_2$-deficient environment (*Figure 3—figure supplement 1C*). Since in both cases *TCS* reporter activity could be restored by sucrose supplementation (*Figure 3K* and *Figure 3—figure supplement 1D*), we concluded that CK signaling output is dependent on the availability of energy metabolites. However, light signaling and photosynthesis together had a much stronger effect on CK output than nutrient availability alone, suggesting that both signals synergize to stimulate CK signaling at the SAM. In contrast, PIN1 localization to the plasma membrane was fully dependent on light perception and could not even be restored by the *cop1* mutation (*Figure 3E,F*).

If CK signaling indeed integrates energy status and light perception, it may be sufficient to activate SAM development. In line with this idea, the importance of CK in light dependent SAM activation had already been demonstrated (*Chory et al., 1994*; *Skylar et al., 2010*) and Yoshida et al. had shown that the application of CK to tomato apices can induce organogenesis in the dark (*Yoshida et al.,2011*). Consistently, etiolated *Arabidopsis* seedlings treated with CK produced leaf

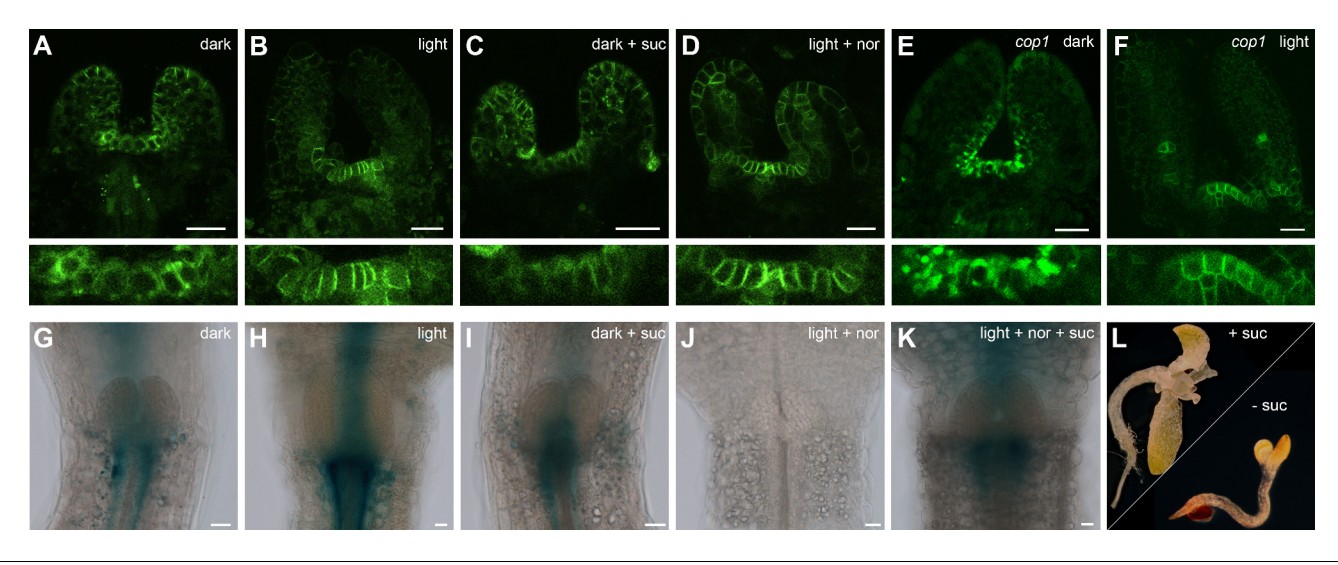

**Figure 3.** Hormonal control of the SAM. (A–F) Confocal images of four day old seedlings expressing *pPIN1:PIN1-GFP* in WT (**A–D**) or *cop1-4* (**E,F**) background under diverse growth conditions. The lower row shows a magnification of the meristem shown in the picture above. (**G–K**) GUS staining of four day old plants expressing *pTCSn:GUS* (light = white light (150 µmol*m$^{-2}$*s$^{-1}$), + suc = 1% sucrose, nor = 5 µM norflurazon, scale bar = 20 µm). (**L**) Wild-type seedlings after 20 days on plates containing CK (75 µM benzyladenine) supplemented with (+) or without (-) sucrose.

The following figure supplement is available for figure 3:

**Figure supplement 1.** *pTCSn:GUS* activity in four day old seedlings grown under different conditions.

like structures even in darkness (*Figure 3L* and *Chory et al. 1994*). However, this developmental transition was strictly dependent on the presence of an external energy source, similar to the behavior of *cop1* mutants and in the absence of sucrose, CK treated seedlings failed to develop leaves in the dark (*Figure 3L*). Thus, our experiments were consistent with CK being an important component of, but not the sole signal for environmental stem cell activation.

Since we had found *WUS* expression to be a much more sensitive readout for SAM activation than organ development, we made use of our reporter system to dissect the role of CK for overcoming stem cell dormancy. Using hormone treatment assays we found that CK alone was sufficient to induce low levels of *WUS* expression even in darkness in line with its known role in SAM regulation (*Gordon et al., 2009*; *Buechel et al., 2010*). Interestingly, we observed the strongest stimulation of reporter activity in plants on sucrose medium, whereas the light response was largely unaffected by CK treatment (*Figure 4A*). These results suggested that application of CK can at least partially replace the perception of light and thus supported a role for CK as a mobile transducer for light signals upstream of *WUS* expression. However, as demonstrated by our results using the *TCS* reporter, in the absence of energy metabolites downstream CK signaling cannot be fully activated, resulting in a lack of organ development. Having established an environment specific role for CK in the initial steps leading up to stem cell activation, we wondered about the mechanism of regulating endogenous hormone levels. We therefore mined the literature for light responsive genes, expressed in the meristem and functionally related to CK metabolism, perception, or signaling. Only *CYTOKININ OXIDASE 5 (CKX5)*, which codes for one of seven homologous cytokinin dehydrogenases involved in CK catabolism in *Arabidopsis* met all criteria (*Frebort et al., 2011*). *CKX5* is a direct transcriptional target of several PHYTOCHROME INTERACTING FACTORs (PIFs) and is highly expressed in etiolated seedlings as well as under shade conditions (*Hornitschek et al., 2012*; *Zhang et al., 2013*; *Pfeiffer et al., 2014*). Similar to *cop1* mutants, also *pifq* mutants form leaves in darkness when supplied with sugars externally, therefore the PIFs are likely to play an important role in suppressing SAM activation in darkness (*Figure 4—figure supplement 1A*). *CKX6*, a close homologue of *CKX5*, had already been described to limit primordia growth in response to shade treatment

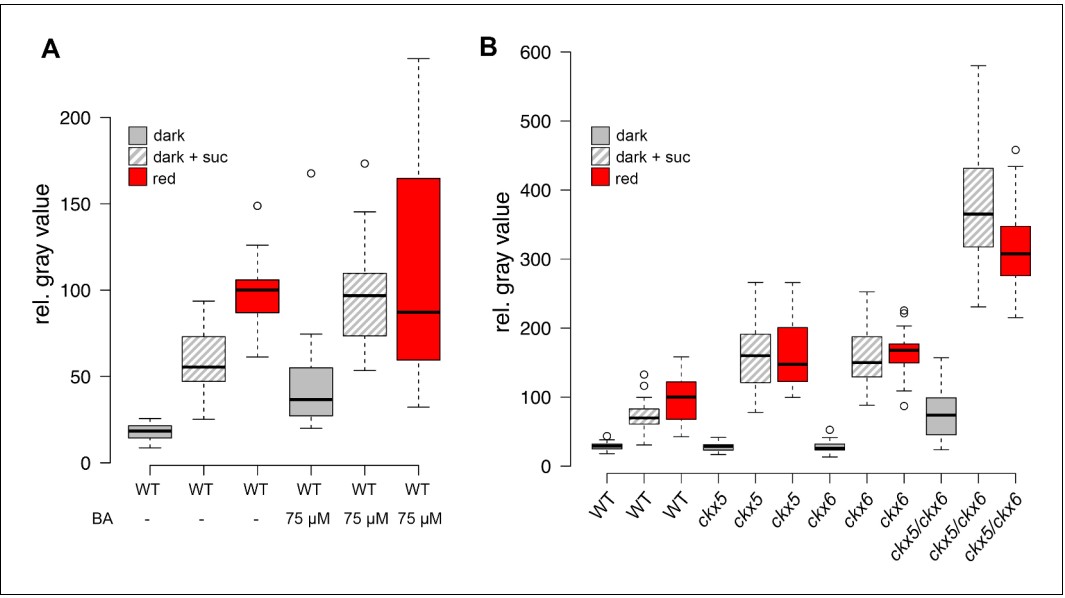

**Figure 4.** Role of cytokinin signaling in *WUS* activation. Quantification of *pWUS:3xVENUS-NLS* expression in four day old seedlings. (**A**) Cytokinin was applied as 75 µM benzyladenine to wild-type (WT). (**B**) *Ckx* mutant lines display enhanced expression of the *pWUS:3xVENUS-NLS* reporter construct (growth conditions: gray = darkness, red = red light (30 µmol*m$^{-2}$*s$^{-1}$), solid box = w/o sucrose, hatched box = 1% sucrose).

The following figure supplement is available for figure 4:

**Figure supplement 1.** Role of *PIF* and *CKX* genes for stem cell activation.

(*Carabelli et al., 2007*) and both *CKX5* and *CKX6* had been shown to be expressed around the shoot apex (*Motyka et al., 2003*; *Bartrina et al., 2011*). When we tested the contribution of *CKX5* and *CKX6* to SAM activation by crossing the single mutants to our reporter line, we found that loss of either did not promote *WUS* activity in the dark as CK treatment. In contrast, the responses to sucrose and light were robustly enhanced (*Figure 4B*), revealing that genetically removing etiolation specific antagonists of CK accumulation is sufficient to replace one of the essential environmental signals, but not both. Because *CKX* genes act partially redundantly (*Bartrina et al., 2011*), we decided to remove *CKX6* in the *ckx5* mutant reporter background by CRISPR/Cas9. Of 27 plants tested in T1, three were homozygous *ckx6* mutants, with either insertions or deletions at the 5′end of the *CKX6* locus leading to a shift in the reading frame (*Figure 4—figure supplement 1E*). Seedlings of all three independent *ckx5/ckx6* mutant reporter lines were analyzed in the T2 generation and showed comparable effects on *WUS* expression (*Figure 4B* includes data from one representative *ckx5/ckx6* line, AP101.9). In contrast to either single *ckx* mutant, *ckx5/ckx6* double mutants showed basal *WUS* reporter activity in darkness, similar to its behavior under CK treatment. Furthermore, we observed a dramatic enhancement of the effect of sucrose on *WUS* activity and also light dependent reporter induction was increased two-fold over either single mutant, which was also detectable by qRT-PCR (*Figure 4—figure supplement 1B*). Based on these results we concluded that CKX5 and CKX6 repress activation of the SAM by degradation of the plant hormone CK in darkness. Interestingly, the level of *WUS* expression in the *ckx5/ckx6* double mutant background was almost comparable to one found in *cop1* mutant plants (*Figure 2B*). However, in striking contrast to *cop1*, the *ckx5/ckx6* double mutant displayed a fully etiolated phenotype in darkness (*Figure 4—figure supplement 1C and D*), demonstrating that stem cell activation was fully uncoupled from photomorphogenesis in these plants. Thus, CK can act as an interregional transmitter of light signals specifically for *WUS* expression, but not for general regulators of photomorphogenic growth, suggesting that the developmental integration of light and metabolic signals might occur downstream of CK and locally at the SAM.

## Mechanisms of metabolic stem cell activation

In addition to light, seedling development and *WUS* expression both showed a strong dependency on sucrose. Since sugars can not only act as energy source, but also as signaling molecules, we first aimed to disentangle these functions for SAM activation. To this end, plants were grown on plates supplemented with equimolar amounts (15 mM) of different sugars with diverse energy content for four days in darkness (*Figure 5A*). While mannitol, a non-metabolizable sugar, did not affect *WUS* activity, *WUS* expression could be observed in etiolated seedlings in the presence of glucose and sucrose, with sucrose being approximately twice as effective as glucose. Conversely, palatinose, a non-metabolizable sugar structurally related to sucrose and able to induce the sugar-dependent bud burst of roses in vitro (*Rabot et al., 2012*), was not sufficient to induce *WUS* (*Figure 5A*). These findings strongly suggested that in the context of stem cell activation, sugars do not act as signaling molecules directly, but rather as energy source. In turn, the metabolic status of the plant seemed to be sensed and translated into appropriate cell behavior by the SAM.

A key sensor of nutrient availability in plants is the TOR (TARGET OF RAPAMYCIN) kinase and photosynthesis-mediated activation of the root meristem has recently been described to be under the regulation of TOR (*Xiong et al., 2013*). Characteristic expression changes that were described in response to glucose-TOR signaling (*Xiong et al., 2013*) and E2Fa overexpression (*Vandepoele et al., 2005*; *López-Juez et al., 2008*), namely affecting genes involved in ribosome biogenesis, protein translation and cell proliferation have also been identified in microarray analyses of shoot apex tissue derived from young seedlings (*López-Juez et al., 2008*). These genes were rapidly and synchronously induced by photosynthetically active light preceding organ growth, which lead us to hypothesize that stem cell activation in the SAM might also be under control of the TOR kinase.

Since mutations in TOR are lethal, we used chemical interference to functionally test its contribution to SAM activation. To efficiently inhibit TOR activity, we applied the ATP-competitive TOR kinase inhibitor AZD-8055 (*Montané and Menand, 2013*) on seedlings grown in liquid culture (*Figure 5B–D*). After an initial germination and growth phase of three days in darkness, sugar, light and inhibitor treatments were applied for another three days followed by microscopic analysis of the seedlings. In line with our hypothesis that TOR is required for energy sensing in the SAM, AZD-8055

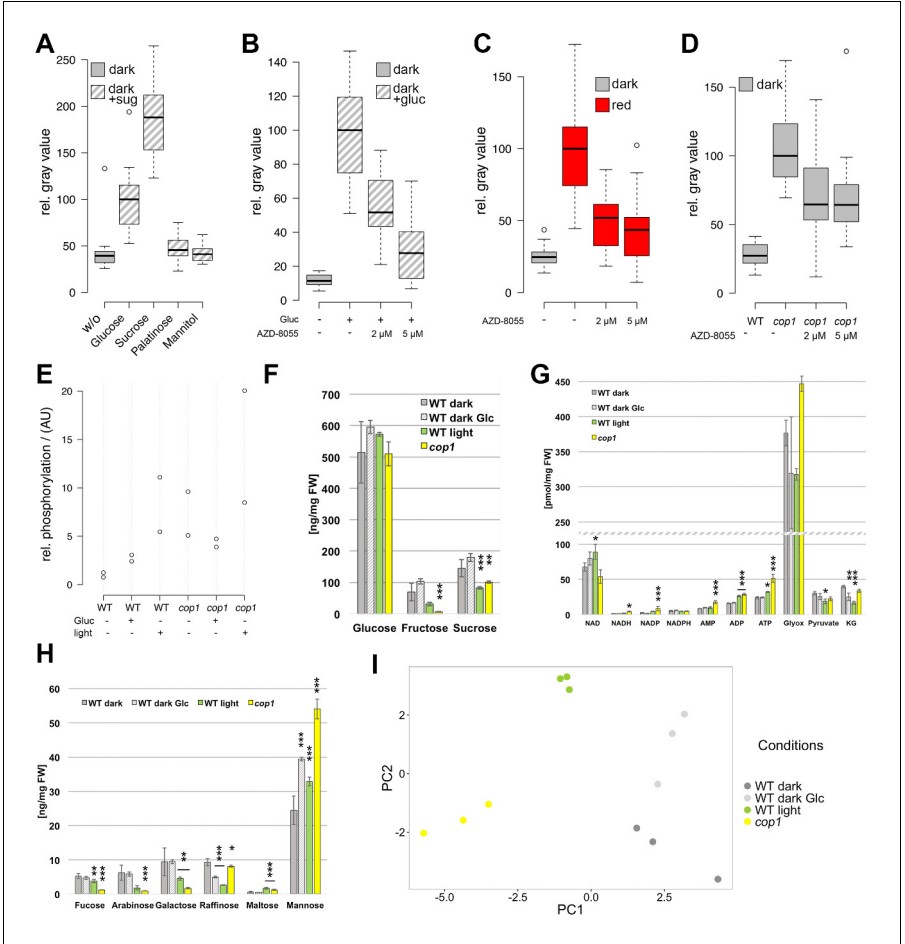

**Figure 5.** The TOR pathway integrates metabolic and light signals upstream of *WUS*. (**A**) Activation of *pWUS:3xVENUS-NLS* in four day old dark-grown seedlings grown with 15 mM of diverse sugars (+sug). (**B–D**) Activation of *pWUS:3xVENUS-NLS* in seedlings grown in liquid culture. (**B**) Effect of AZD-8055 TOR inhibitor on glucose treatment. (**C**) Effect of AZD-8055 TOR inhibitor on light treatment. (**D**) Effect of AZD-8055 TOR inhibitor on *cop1-4* mutant seedlings. (growth conditions: gray = darkness, red = red light (30 $\mu$mol*m$^{-2}$*s$^{-1}$), solid box = w/o sugars added, hatched box = with sugar) (**E**) Quantification of S6K phosphorylation relative to total S6K levels based on western blots shown in *Figure 5—figure supplement 1B*. (**F–H**) Metabolite measurements from four day old seedlings. Gray bars represent untreated seedlings; hatched bars represent 24 hr treatment with 15 mM glucose; green bars represent 24 hr light treatment (60 $\mu$mol*m$^{-2}$*s$^{-1}$ of red and blue light each); yellow bars represent *cop1-4* mutant seedlings. Error bars show standard error of the mean; asterixs indicate significance tested by unpaired two-tailed oneway ANOVA Student-Newman-Keuls test: *p<0.05, **p<0.01, ***p<0.001; Glyox = glyoxylate; KG = ketoglutarate. (**I**) PCA based on metabolite measurements shown in *Figure 5F–H* and *Figure 5—figure supplement 1D,E*.

The following figure supplement is available for figure 5:

**Figure supplement 1.** The TOR pathway is activated by light signal transduction.

inhibited glucose-induced *WUS* expression in a dose dependent manner (*Figure 5B*). Already 2–5 $\mu$M of AZD-8055 resulted in a considerable repression of *WUS* promoter activity when compared to mock controls, and almost completely suppressed the positive effect of glucose.

## TOR acts as an integrator of light and metabolic signals for stem cell activation

Since TOR kinase has such a central function in the regulation of growth and development, we wondered whether besides the well documented nutrient availability pathway other stimuli, such as light signaling, might also depend on TOR activity. Using the same experimental setup as described above for glucose we indeed found that light-induced *WUS* expression was efficiently inhibited by AZD-8055 (*Figure 5C*). This suggested that TOR kinase might play a specific role in light signaling independent of energy production, since the plants were exposed to a rather low light intensity of $30 \ \mu mol*m^{-2}*s^{-1}$. To further test this, we analyzed TOR dependency in the absence of photosynthesis and found that light dependent *WUS* expression was repressed by the TOR inhibitor even under norflurazon treatment (*Figure 5—figure supplement 1A*). Consistently, even genetic activation of the light signaling pathway through the *cop1* mutation in the absence of a physiological stimulus was sensitive to TOR inhibition by AZD-8055 (*Figure 5D*).

Since we could not exclude that our observations were caused by unspecific side effects of the AZD-8055 inhibitor, we wanted to monitor TOR activity in response to sugars and light directly. Working under the hypothesis that TOR activity should correlate with energy and light signals if the inhibitor results were meaningful, we quantified the phosphorylation level of the direct TOR target S6K by a phospho-specific antibody (*Figure 5E* and *Figure 5—figure supplement 1B*). After transfer of three-day-old etiolated seedlings to either light ($60 \ \mu mol*m^{-2}*s^{-1}$ of blue and red light each) or 15 mM glucose for 24 hr, we detected a substantial increase in phosphorylation of S6K compared to seedlings that were kept in darkness (*Figure 5E*). Surprisingly, light treatment was even more efficient in activating TOR kinase than supplying plants with external glucose. Consistently, *cop1-4* mutant seedlings displayed high S6K phosphorylation levels under all conditions, including the dark-grown control. Also in this case we were able to detect a similar activation of the TOR pathway in other *cop1* alleles (*Figure 5—figure supplement 1C*), supporting our previous observation that the TOR pathway is repressed by the negative light signaling component COP1 in darkness. Taken together, TOR kinase seems to play a central and so far underestimated role not only for metabolic, but also for light dependent activation of *WUS* and ultimately stem cells in the SAM.

## Environment dependent metabolic reprogramming

After having shown that light and energy status converge to activate stem cells, we wondered about the underlying metabolic changes in response to these environmental cues. We were most interested to analyze metabolic re-programming in response to light signaling because we hypothesized that TOR could be indirectly activated following at least two scenarios: First, since in the wild light signaling and photosynthesis usually go hand in hand, we were wondering whether signaling alone would trigger a metabolic shift in expectation of photosynthesis derived energy metabolites. And second, since *COP1* regulates more than 20% of the *Arabidopsis* genome (*Ma et al., 2002*), we speculated that this could also extend to the activity of metabolic enzymes. In both cases the metabolic state of the seedlings, especially with respect to glucose levels, could be affected by light signaling and this in turn could indirectly trigger TOR. We therefore analyzed metabolites in seedlings that were subjected to the same experimental workflow as for studying TOR activity described above, namely wild-type and *cop1-4* seedlings grown in darkness compared to seedlings treated with either 15 mM glucose or light for 24 hr before harvest (*Figure 5F–H*; *Figure 5—figure supplement 1D–F*; *Supplementary file 1*). A Principal Component Analysis (PCA) plot clearly identified light signaling triggered by the *cop1* mutation (PC1) and energy availability caused by glucose supplementation (PC2) as the two major components in our sample set, with light treated samples being influenced by both components. Interestingly, glucose treatment only had a mild effect on seedling metabolism. We detected slight increases of glucose, fructose and sucrose as well as a clear increase of mannose levels in glucose treated seedlings, while raffinose levels were reduced (*Figure 5F,H*). The fact that under these conditions TOR activity is markedly increased and *WUS* expression is robustly detectable demonstrates that the energy sensing machinery must be exquisitely sensitive to small changes in metabolite content, most likely glucose. To our surprise, *cop1* and light treated seedlings showed similar metabolic profiles that were clearly different from the glucose treated samples. Since *cop1* seedlings were grown in darkness, the metabolic re-programming in *cop1* and light treated seedlings is likely caused by light signaling dependent transcriptional regulation of metabolic

enzymes, rather than simple photosynthesis dependent accumulation of sugars. We detected a clear reduction of fructose and sucrose levels and a decrease in the cell wall monosaccharides fucose, arabinose and galactose in both samples (*Figure 5H*). In addition, ADP and ATP levels were slightly increased by light and *cop1* mutation (*Figure 5G*). *Cop1* seedlings also displayed an accumulation of glyoxylate, an intermediate of the glyoxylate cycle that allows plants to use lipids as carbon source (*Figure 5H*). Another noteworthy exception from the similarity between *cop1* and light treated seedlings was glucose. Importantly, only the light- and glucose-treated samples, but not *cop1* seedlings showed an increase in glucose levels that might be responsible for the observed activation of TOR kinase. However, since it is not known how the TOR complex is activated in plants, we cannot exclude that the obvious metabolic re-programming indirectly triggered TOR kinase activity in *cop1* seedlings. Nevertheless, since *cop1* seedlings showed an overall reduction of energy rich metabolites, it seemed unlikely that the nutrient state was exclusively responsible for activation of the TOR pathway in the *cop1* background.

Interestingly, we saw a significant increase in the levels of specific amino acids and their derivatives especially in *cop1* (Glu, Gln, His, Arg, ornithin, spermidine and citrulline) that were all previously described to be reduced in *amiR-tor* seedlings (*Figure 5—figure supplement 1F* and *Supplementary file 1*). Conversely, time amino acids whose levels were reported to be increased upon TOR repression (Thr, Tyr, Val, Ile and Leu) were unchanged or reduced in *cop1* (*Caldana et al., 2013*). The inverse correlation of amino acid levels in *cop1* and *amiR-tor* lines was not apparent in the other samples, suggesting that the permanent de-regulation of the TOR pathway during either *cop1* or *amiR-tor* seedling development strongly affected the metabolome while the short term nature of the glucose or light treatments was insufficient for such a profound change.

## Discussion

The life of a plant begins in many cases with skotomorphogenesis, which is characterized by the elongation of the hypocotyl, formation of an apical hook, unfolded cotyledons and the dormancy of the SAM. While most characteristics of skotomorphogenesis can be revoked by the perception of light through the photoreceptors alone, we showed that activation of the SAM in addition requires the presence of energy metabolites. Thus, for stem cell activation, light not only acts as a signal, but also needs to fuel the photosynthetic apparatus to produce sugars and despite the dual role of a single environmental factor, both inputs are sensed independently. Amazingly, it is not stem cell fate that is dependent on these signals, but rather the expression of *WUS*, which defines the niche and at the same time acts as a mobile stem cell activator. Thus, stem cell fate as defined by *CLV3* promoter activity can exist without WUS in a physiologically and developmentally relevant setting, strongly suggesting that WUS is not the primary stimulating input for *CLV3* expression.

For the sensing, transmission and integration of light and metabolic signals by the stem cell system, evolution seems to have co-opted well studied regulators into a novel and so far unsuspected regulatory network. On the one hand, the roles of the phyB and cry photoreceptors for stem cell activation are fully in line with text book photomorphogenesis, on the other hand phyA exhibits novel positive and negative functions in far-red light regulation of *WUS* expression, which point to a re-wiring of this core component of light signaling. Similarly, in light signal transduction, the etiolation regulator COP1 plays a prominent part and *cop1* mutations can substitute for light in essentially all experiments, apart from PIN1 membrane localization. In contrast, HY5, another core component of the photomorphogenesis network, does not seem to have any apparent role in light dependent stem cell activation. This is even more striking when taking into account the recently discovered role in shoot-to-root signaling of HY5 (*Chen et al., 2016*), and our observation that the meristem region itself does not possess the competence to perceive and process light signals. However, we cannot exclude that the HY5 homolog HYH with partially overlapping function but higher expression level in the shoot might mask the effect of *hy5* on *WUS* expression (*Holm et al., 2002*; *Sibout et al., 2006*). Since *hy5/hyh* mutants showed deficiencies in development of the first leaves future experiments analyzing such a double mutant in the context of our double reporter are required.

Interregional transmission of the light signal seems to depend on CK and with the cytokinin dehydrogenases *CKX5* and *CKX6*, we identified two potent regulators of meristem activity that are highly responsive to environmental light conditions. Both are direct targets of PIFs and have already been shown to accumulate in etiolated seedlings (*CKX5*) and shade conditions (*CKX6*), respectively

(*Carabelli et al., 2007*; *Pfeiffer et al., 2014*), which ultimately leads to an inactivation of cytokinin under unfavorable light conditions. Under open sun light PIFs are degraded, thus liberating cytokinin from CKX mediated degradation, which in turn results in the activation of stem cells. *CKX5* is expressed in the rib zone below the stem cell niche and *CKX6* additionally in the vasculature (*Motyka et al., 2003*; *Bartrina et al., 2011*), which might explain the surprisingly high efficiency of our *pSUC2:PHYB Y276H* line in activating *WUS* expression (*Figure 2—figure supplement 1G*). We did not specifically activate light signaling in the rib zone below the stem cell niche in our experiments and thus the possibility remains that light perception in the immediate vicinity to the meristem affects stem cell activity by short range CK signaling.

In parallel to light signals, *WUS* expression was strongly responsive to energy availability. The effect of sucrose on *cop1* and *ckx5/ckx6* double mutants was much stronger than the cumulative effect of light and sucrose treatments (*Figures 2B* and *4B*). Also the activity of the CK output reporter *pTCSn:GUS* strictly depended on sucrose, but could be further stimulated by light, suggesting that CK signaling mainly acts to transmit light signals, but that elevation of CK levels genetically or by treatment are insufficient to elicit the full developmental response. Based on these results we suggest that light enhances CK levels by reducing the expression of *CKX* genes, however we cannot rule out other explanations, such as stimulation of CK biosynthesis.

Interestingly, the WUS paralog *WOX9/STIMPY (STIP)* also plays an important role in light, sugar and CK crosstalk at the shoot apex. Meristems of weaker *stip* mutants arrest at seedling stage but can be rescued by addition of sucrose to the medium (*Wu et al., 2005*). *STIP* was further shown to integrate CK signals at the meristem and *STIP* over-expression can partially overcome the deficits of CK perception mutants (*Skylar et al., 2010*). In contrast to *WUS*, however, sugar acts downstream of *STIMPY*, and it would be interesting to investigate whether the same is true for light signals.

Our experiments suggested that light and metabolic signals converge downstream of CK and locally at the SAM and consistently, we identified the TOR kinase to be an integrator of both signaling pathways. Earlier reports had shown that glucose-TOR signaling regulates photosynthesis-driven activation of the root meristem (*Xiong et al., 2013*) and we demonstrated here that also shoot stem cell activation by metabolizable sugars is dependent on TOR kinase. Intriguingly, TOR activity was not only required for the response to energy availability, but also for *WUS* stimulation by light. This activity was independent of photosynthesis, because the TOR pathway was also activated under norflurazon treatment, in the absence of $CO_2$ or the suppression of COP1 function. The direct regulation of TOR kinase by light signaling elegantly explains the recently discovered impact of phytochromes on the metabolic state of the plant (*Yang et al., 2016*), but also represents another striking example of a so far undiscovered re-wiring of a core regulatory component within the stem cell network. Consistently, upstream regulators of TOR kinase described in other organisms are not well conserved in plants (*Dobrenel et al., 2016*) and thus evolution seems to have found alternative mechanisms to activate the TOR pathway in a highly context dependent manner. Coupling light and energy sensing via TOR could help plants to prepare for the dramatic developmental transition from skotomorphogenesis to photomorphogenesis, which involves transcriptional re-programming of almost a quarter of the genome (*López-Juez et al., 2008*). Light dependent activation of TOR kinase could allow etiolated seedlings to build up the photosynthetic apparatus, initiate ribosome biogenesis and prime stem cells via the expression of *WUS* to efficiently shift gear towards organogenic growth and development, once energy becomes available.

## Material and methods

### Cloning
#### Double reporter
The *pCLV3:mCHERRY* (pMD149) construct was obtained by LR reactions of the *pDONR221-mCHERRY-NLS* plasmid (pFK143) with pFK317, a pGreen-IIS (*Hellens et al., 2000*; *Mathieu et al., 2007*) based binary vector with *CLV3* 1.4 kb promoter and 1.2 kb terminator sequences and kanamycin resistance cassette. The *pWUS:3xVENUS-NLS* construct (pTS81) was generated accordingly, using a pGREEN-IIS based binary vector with a 4.4 kb genomic *WUS* promoter upstream of the ATG and 2.8 kb *WUS* terminator downstream of the stop codon (pFK398) and a Basta resistance cassette. In both constructs the N7 NLS (*Daum et al., 2014*) was used.

To produce *ckx6* mutants by CRISPR-Cas9 the two gRNAs GCATGGTTCTTTTCCTGAGG and GAAGCTGCAGGTCTACAGTG targeting the *CKX6* locus were inserted in the plasmid pHEE401E as described by the authors (*Wang et al., 2015*) to generate pAP101.

All other constructs are based on the GreenGate cloning system (*Lampropoulos et al., 2013*). Details about the cloning of the modules and assembled plasmids for plant transformation can be found in *Supplementary file 2*.

## Plant material

All used plant lines were in the Col-0 background. *Arabidopsis* Col-0 was transformed by floral dip (*Clough and Bent, 1998*) using the *A. tumefaciens* strain ASE (pSOUP+) carrying the pGREEN-IIS and GreenGate based plasmids. The *Agrobacteria* strain GV3101 was used for the transformation of plants with the plasmid pAP101.

To select for the presence of the corresponding resistance markers, plants were grown on plates supplemented with 5 mM D-Ala, 20 µg/ml Hygromycin B or 50 µg/ml Kanamycin or grown on soil and sprayed with the herbicides Inspire (Syngenta Agro AG, Dielsdorf, Switzerland) (7.6 µl/l) (*Rausenberger et al., 2011*) or Basta (0.02%) 1 week after germination.

The plasmids pMD149 and pTS81 were used to generate the reporter lines *pCLV3:mCHERRY-NLS* and *pWUS:3xVENUS-NLS*, respectively. The double reporter line is comprised of a cross of both lines. The WUS-GFP rescue line (*pWUS:WUS-linker-GFP* in *wus* mutant background [*Daum et al., 2014*]) was also crossed to the *pCLV3:mCHERRY-NLS* line. Both crossed lines were homozygous for all loci.

The mutant *cop1-4* (*McNellis et al., 1994*) was crossed to *pPIN1:PIN1-GFP* (*Benková et al., 2003*) and the double reporter line was crossed to the mutants *cop1-4* (*McNellis et al., 1994*), *phyA-211* (*Reed et al., 1994*), *phyB-9* (*Reed et al., 1993*), *cry1-304/cry2-1* (*Mockler et al., 2003*), *hy5* (SALK_096651C), *ckx5-1* (SALK_064309) and *ckx6-2* (SALK_070071) (*Bartrina et al., 2011*). All of these crossed lines used in the manuscript were homozygous mutants and screened to be also homozygous for carrying the *pWUS:3xVENUS-NLS* construct. Additional *cop1* alleles, *cop1-6* (*McNellis et al., 1994*) and *cop1-19* (*Favory et al., 2009*), were used for qRT-PCR and the TOR activity assay.

A homozygous double reporter/*ckx5* line was transformed with the CRISPR/CAS9 plasmid pAP101 to create a *ckx5/ckx6* double mutant in the double reporter background. Genomic DNA of T1 plants was PCR-amplified using the oligos A05465 (ATCAAAAACCCTTTTCCATCCT) and A05466 (AGCCAACTTAAAGGCTATGCAG) and the PCR product was digested with *Eco81I* to screen for homozygous *ckx6* mutants at the locus of the first gRNA. The genomic region of T1 plants that produced undigested PCR fragments was amplified with the oligos A05465 and A05468 (ACTTGAGGG TCTCATGCAAAAT), and sequenced after subcloning the PCR product into pGEM-T Easy (Promega, Madison, WI)) to confirm the mutation of the *CKX6* locus. T2 plants of homozygous T1 mutants were used in this manuscript.

## Growth conditions

Seeds were sterilized with 70% ethanol and 0.1% Triton for 10 min and afterwards washed twice with autoclaved water. All seeds were imbibed in water for three days at 4°C in darkness before plating 30–40 seeds on 0.5x MS (Duchefa, Haarlem, The Netherlands), 0.8% Phytoagar in vented petri dishes that were sealed with micropore tape (3 M, Two Harbors, MN). Germination was induced by 150 µmol*m$^{-2}$*s$^{-1}$ of white light for 6 hr. Afterwards plants were either kept in white light, transferred to darkness or to LED cabinets equipped with red (673 nm), far-red (745 nm) and blue (471 nm) LEDs (floralLEDs StarterKit 2, CLF Plant Climatics, Wertingen, Germany). Unless otherwise noted, constant red, far-red or blue light was applied with an intensity of 30 µmol*m$^{-2}$*s$^{-1}$. All white light treatments were carried out at 150 µmol*m$^{-2}$*s$^{-1}$ of fluorescent white light with a 16-h-light/8-h-dark cycle. Starting with the light induction of germination, plants were kept at 21–22°C. 0.5x MS plates were supplemented with 1% (30 mM) sucrose or 15 mM glucose only if mentioned explicitly. Growth in a CO$_2$-deficient environment was accomplished by growing unsealed vented petri dishes in a sealed plastic bag with 5 g NaOH and 5 g CaO (*Kircher and Schopfer, 2012*).

Four day old seedlings that were harvested for protein extracts and metabolite measurements were grown vertically on top of 100 µm nylon meshes (nitex 03/100–44, Sefar, Heiden, Switzerland).

For the glucose treatment in these experiments seedlings were transferred with the mesh to plates containing 15 mM glucose. Light treatments entailed irradiation with 60 µmol*m$^{-2}$*s$^{-1}$ of blue and red light each. Both treatments were started 24 hr before and continued until the harvest of the material. To exclude ungerminated seeds and empty seed shells form the metabolite measurements, only above root tissue was harvested. All seedlings were rinsed with distilled water prior to harvest.

## Liquid culture

About 30–40 seeds, that were imbibed as described above, were sown in 3 ml 0.5x MS in petri dishes of 35 mm diameter. Plants were kept in darkness for three days after the induction of germination by 6 hr light treatment. The medium of three day old etiolated seedlings was supplemented with 15 mM glucose and/or 0.5–2 µM AZD-8055 (Santa Cruz Biotechnology, Dallas, TX). Stock solutions of 1000x concentrated AZD-8055 were diluted in DMSO, therefore control plants were mock treated with the same volume of DMSO.

## In situ hybridization

A detailed protocol of the in situ hybridization procedure was provided previously (*Medzihradszky et al., 2014*).

## RNA extraction and qRT-PCR

Total RNA was extracted from 100 mg seven day old *Arabidopsis* seedlings with the Plant RNA Purification Reagent (Invitrogen, Carlsbad, CA) according to the instructions of the manufacturer, digested with TURBO DNAse (Ambion/ Thermo Fisher, Waltham, MA) and purified with RNeasy Mini Kit (Quiagen, Hilden, Germany). Equal amounts of RNA were used for oligo dT primed cDNA synthesis with the RevertAid First Strand cDNA Synthesis Kit (Thermo Fisher, Waltham, MA). The qPCR reaction was set up using the SG qPCR Master Mix (EURx, Gdansk, Poland) and run on a Chromo4 Real-Time PCR System (Bio-Rad, Hercules, CA) with technical duplicates each. The relative expression levels were calculated using the ddCt method with *PP2A* expression as a reference. Results shown are the means of 2 independent biological replicates. The following oligos were used: PP2A: A01067: TAA CGT GGC CAA AAT GAT GC and A01068: GTT CTC CAC AAC CGC TTG GT; WUS: A00317: TTA TGA TGG CGG CTA ACG AT and A00318: TTC AGT ACC TGA GCT TGC ATG; PHYB total: A05986: AGC AAA TGG CTG ATG GAT TC and A05987: GCT TGT CCA CCT GCT GCT AT; PHYB 3'UTR: A05984: GCG ACC ATT GTC AAC TGC TA and A05985: CTC CGA CGT CGT TAG ACA CA.

## Histochemical GUS staining

Four day old seedlings were harvested in 90% acetone and incubated at −20°C for at least 1 hr. Seedlings were washed with PBS and incubated in substrate buffer (1x PBS (pH 7.0), 1 mM K$_3$Fe(III)(CN)$_6$, 0.5 mM K$_4$Fe(II)(CN)$_6$, 1 mM EDTA, 1% Triton X-100, 1 mg/ml X-gluc) at 22°C over night. After staining, the seedlings were incubated with 60% and subsequently in 95% ethanol to remove chlorophyll.

## Microscopy and fluorescence quantification

To image the fluorescent reporter activities in the SAM, seedlings were split in half by pulling one cotyledon away from the SAM with forceps. The exposed meristem was imaged with a Zeiss Imager M1, the Plan-APOCHROMAT 20x/0.8 objective (Zeiss, Oberkochen, Germany) and YFP- and mCHERRY-specific filter sets. For the quantification of VENUS and mCHERRY signal intensities the settings for the intensity of the fluorescent lamp and exposure times were unchanged for each channel. 16-bit B/W pictures of at least 20 SAMs per sample were analyzed by FIJI (*Schindelin et al., 2012*), using the background subtraction (100 pixel rolling ball radius) prior to measuring the mean gray value of a circular area surrounding the SAM with a diameter of 51 µm (100 pixels) for *WUS* and 41 µm (80 pixles) diameter for *CLV3*. Quantifications in each figure were normalized to the median of the fluorescence levels of wild-type plants grown in red light for four days. Only exception: in *Figure 5A and B* we used the glucose treated plants as a reference and in *Figure 5D* the *cop1* mutant plants (second box in each box plot). For all these experiments plants of one experimental

set were always grown and analyzed in parallel to the untreated (dark-grown) and the corresponding reference sample.

In situ sections were analyzed with the same microscope and a 40x/0.95 Plan-APOCHROMAT objective (Zeiss, Oberkochen, Germany). Equipment and settings used for confocal microscopy was described earlier (*Daum et al., 2014*).

## TOR activity assay

Proteins were extracted from 50 mg materials in 250 µl 2x Laemmli buffer (0.25 mM Tris-HCL pH 6.8, 8% SDS, 5% ß-mercaptoethanol, 20% glycerol) supplemented with 1.5% phosphatase inhibitor cocktail 2 (Sigma-Aldrich, St. Louis, MO). After adding extraction buffer, samples were briefly mixed and heated at 95°C for 10 min. Cellular debris was removed by two centrifugation steps (10 min, 14,000 rpm, 4°C). 20 µg protein were separated on a 10% SDS gel and transferred to PVDF membrane. Phospho-p70 S6 kinase (Thr(P)-389) polyclonal antibody (No.9205, Cell Signaling Technology, Cambridge, UK) was used to detect S6K phosphorylation. S6K1/2 antibody (AS12-1855, Agrisera AB, Vännäs, Sweden) was used to detect total S6K1 and S6K2.

## Metabolite measurements

Three biological replicates were harvested for and analyzed as described (*Poschet et al., 2011*) by the Metabolomics Core Technology Platform at the University of Heidelberg. PCA was performed with statistical language R (version 3.3.1). For the analysis all metabolite data except the amino acid measurements were used.

## Acknowledgements

We thank Cornelia Klose and Roman Ulm for sharing photoreceptor and light signaling mutants, Andreas Hiltbrunner for the plasmid for mPPO and Inspire, Qi-Jun Chen for providing the CRISPR-Cas9 plasmids and the Metabolomics Core Technology Platform at the University of Heidelberg. This work was supported by the ERC grant (#282139) 'StemCellAdapt' to JL and the German research foundation (DFG) grants SFB873 to JL and grants HE 1848/15-1, WI 3560/1-1 and SFB 1036 to RH and MW. AP is indebted to the Baden-Württemberg Stiftung for the financial support of this research project by the Eliteprogramme for Postdocs.

## Additional information

### Funding

| Funder | Grant reference number | Author |
| --- | --- | --- |
| Baden-Württemberg Stiftung | Eliteprogramme for Postdocs | Anne Pfeiffer |
| Deutsche Forschungsgemeinschaft | CellNetworks Cluster of Excellence | Anne Pfeiffer |
| Deutsche Forschungsgemeinschaft | HE 1848/15-1 | Markus Wirtz Rüdiger Hell |
| Deutsche Forschungsgemeinschaft | WI 3560/1-1 | Markus Wirtz Rüdiger Hell |
| Deutsche Forschungsgemeinschaft | SFB 1036 | Markus Wirtz Rüdiger Hell |
| European Research Council | Starting Grant 282139, StemCellAdapt | Jan U Lohmann |
| Deutsche Forschungsgemeinschaft | SFB 873 | Jan U Lohmann |

The funders had no role in study design, data collection and interpretation, or the decision to submit the work for publication.

## Author contributions

AP, DJ, Conception and design, Acquisition of data, Analysis and interpretation of data, Drafting or revising the article; YD, AM, SS, Acquisition of data, Analysis and interpretation of data, Drafting or revising the article; GD, Acquisition of data, Drafting or revising the article, Contributed unpublished essential data or reagents; TS, JF, TL, MS, Drafting or revising the article, Contributed unpublished essential data or reagents; ER, MW, RH, Analysis and interpretation of data, Drafting or revising the article; JUL, Conception and design, Analysis and interpretation of data, Drafting or revising the article

## Author ORCIDs

Rüdiger Hell, http://orcid.org/0000-0002-6238-4818
Jan U Lohmann, http://orcid.org/0000-0003-3667-187X

## Additional files

### Supplementary files

• Supplementary file 1. Results of metabolomic analysis.

• Supplementary file 2. Description of constructs and oligonucleotides used in this study. To generate new Green Gate module plasmids PCR was carried out with Phusion polymerase and the oligos listed on Arabidopsis genomic DNA, cDNA or diluted plasmids. PCR product were gel purified, digested with *Eco31I* and ligated in empty entry vectors that were *Eco31I* digested and gel purified. All modules were checked by sequencing.

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
