## [Decision Letter]

Thank you for submitting your article "Integration of light and metabolic signals for stem cell activation at the shoot apical meristem" for consideration by *eLife*. Your article has been favorably evaluated by Ian Baldwin (Senior editor) and three reviewers, one of whom is a member of our Board of Reviewing Editors. The reviewers have opted to remain anonymous.

The reviewers have discussed the reviews with one another and the Reviewing Editor has drafted this decision to help you prepare a revised submission.

All reviewers agree that your work is of general interest and in principle support publication. However, they also agree that a few issues have to be addressed before the work can be accepted. In your revision, please pay special attention to the following points:

1) The reviewers are worried about the validity of some of the pharmacological experiments because of possible indirect or side effects of the drugs (in particular of norfluorazon and lincomycin). Therefore, they believe that your paper would be substantially strengthened if you could repeat the basic introductory experiments (Figure 1 and Figure 2) in a CO_2_-free atmosphere.

2) The reviewers also have some concerns regarding the plant material, especially with regard to the *cop1* allele used. They ask that some of the basic experiments are verified with a *cop1* null allele.

3) The reviewers acknowledge the use of reporters as an output for *CLV3/WUS* activity, but also wonder whether some of the basic observations could be confirmed by qPCR analysis of the endogenous genes?

Please find the reviewer comments attached below for more details, and please respond to them to the best of your capacity.

Reviewer #1:

This manuscript from the Lohmann lab and colleagues presents a complex set of data, which indicate that SAM activation is both light- and energy-status dependent in *Arabidopsis*. Overall, the data appear solid and are presented in a transparent manner, and the conclusions are largely justified from my point of view. However, the authors' conclusions strongly hinge on the quantification of two reporters under different conditions or in different genetic backgrounds, and I believe some improvements are required to make the manuscript acceptable for publication.

1) I have one major concern with respect to the genetic material. Several analyses are based on experiments with a mutant in COP1, a central negative regulator of photomorphogenesis. Understandably, the authors use one of the viable *cop1* alleles in their work, the *cop1-4* mutant. However, within the *cop1* allelic series, this allele has always been the most mysterious. Unlike true null alleles, *cop1-4* produces a truncated protein that comprises both the RING finger and coiled coil domains, yet alleles that retain these domains as well but are truncated further C-terminal or carry certain point mutations in the WD40 repeat domain also have a strong phenotype.

Therefore, I believe it is important to check whether the results obtained with *cop1-4* are representative. I understand that it would be too much to ask to repeat reporter experiments in another *cop1* background, but I think the authors could check the principal physiological responses in true *cop1* null alleles, which should be easily available. In particular, the authors could repeat the experiment of Figure 1 and Figure 5. They could also try to detect the effect of *cop1* on *WUS* expression by qPCR instead of a reporter, the difference appears strong enough although it is restricted to few cells (Figure 2).

2) Another genetic concern: I believe the authors are formally right to conclude that HY5 has no role in light-dependent stem cell activation, however it is important to remember that unlike *cop1*, none of the COP1-target mutants has a phenotype that is as absolute skotomorphogenic as *cop1* is photomorphogenic. If their conclusions with respect to *cop1* are right, multiple mutants combining various COP1 targets should eventually show the pertinent defect. Among others, the *hy5 hyh* double mutant comes to mind. In the shoot, HYH is stronger expressed than HY5, and *hy5 hyh* double mutants have been reported to have meristematic defects, like "delayed" organ formation. I believe the authors should check light-dependent SAM activation at least in *hyh* single and *hy5 hyh* double mutants, and maybe also pertinent triple mutants with additional factors.

3) With respect to responses illustrated by images only, like *cop1* shoot formation on +/- suc, are those responses fully penetrant? If so, this should be explicitly stated, if not, proportions clearly indicated.

4) One of the two ML1 transgenic lines deviates from the highlighted phenotype (Figure 2—figure supplement 1). Why is that? Is it an expression problem? As I understand, the *phyB* mutant was not tagged? Maybe the authors can present evidence from additional lines and/or correlation with expression levels. As is, their statement regarding ML1 is misleading, because one out of two lines shows the other phenotype, so it's a 50:50 choice.

5) With regard to the cytokinin experiments, and the sometimes inconclusive contribution, I wonder whether the actual cytokinin used is of importance? Given the variety of cytokinins and their reported shoot- and root-specific activities, the authors might find a "better" cytokinin, which might have a more clear effect on WUS expression.

6) Because of the importance of the fluorescence measurements, I would like the authors to confirm that samples analyzed for one graph were grown in parallel and analyzed with identical settings.

Reviewer #2:

This is a very interesting manuscript exploring how light and metabolic signals activate the SAM. The SAM is dormant in etiolated seedlings and gets activated upon light perception. As shown previously the SAM of etiolated *cop1* seedlings is activated by adding sugar thereby leading to the formation of leaves (even flowers) in darkness. This suggests that both light signals (constitutively on in *cop1*) and metabolic signals cooperate to enable SAM activation. The authors explore the molecular mechanisms underlying this activation. The authors show that the *CLV3* expression domain is rather constant across conditions while *WUS* expression is induced by light. A number of experiments indicate that this light activation comprises both light and metabolic signals. TOR is implicated in the metabolic regulation of *WUS* expression while the phytohormone CK is involved in light regulation of *WUS* expression. This paper should be of broad interested for developmental biologists. Below I list a few issues that the authors should clarify.

1) Based on WUS expression in *cop1* +/- sucrose and WT +/- light and the use of NF and Linc (Figure 2), the authors conclude that both light and metabolic signals contribute to *WUS* expression. How can we be sure that NF and Linc were really effective in the experiment shown in 2A? More generally, if at all possible it would more elegant/convincing to show the role of light on the production of sugars by performing at least one experiment in a CO_2_ free environment (see for example Kircher and Schopfer 2012). This would not lead to the rather drastic effects of these drugs on chloroplasts. In principle the effect of CO_2_-free air could be complemented by adding sucrose. Haydon et al., 2013 also complemented the NF effect by adding sucrose which to some extent rules out non-specific effects of NF.

2) The role of CK is potentially quite complex and I'm not sure that the current data allows drawing a clear conclusion. The authors propose that CK promote *WUS* expression by transmitting the light signal (e.g. subsection “Mechanisms of hormonal stem cell activation”, end of last paragraph). In Figure 2 one can observe that light-grown plants treated with NF still show strong *WUS* expression. However, on Figure 3 light grown seedlings treated with NF show absolutely no TCS:GUS signal, indicating no CK activity. How then could such seedlings express *WUS* (Figure 2) given that the metabolic signal should be gone (in the presence of NF) and TCS:GUS is also gone? A second problem with the model is the role of light-regulated *CKX5/6* expression. The authors propose that this is how light regulates the CK function in this process. However, light regulation of *WUS* expression is just as robust in the *ckx5/ckx6* double mutant than in the WT (4B). It therefore rather appears that these genes are involved in basal regulation of *WUS* expression and not in light regulation. It would actually be interesting to test this by adding sucrose to the double mutant and look for signs of leaf development in darkness rather than hypocotyl elongation as shown in supplementary Figure 4.

3) A potential difficulty with the role of CK in this process and the use of drugs like NF is that CK biosynthesis requires plastids and chloroplast development is promoted by CK. Not sure how this can be untangled but it should be discussed.

4) All figure legends should be more explicit and provide detailed information about the exact light conditions (intensity, color), presence of sucrose and numbers of seedlings used per experiment. For example, I don't see which data correspond to + sucrose (hatched box) in 2A.

Reviewer #3:

This interesting manuscript investigates the regulation of the stem cell inducer *WUSCHEL (WUS*) and of it repressor CLAVATA3 (*Clv3*) by metabolic, hormonal and light signals using fluorescent markers driven by the promoters of these genes. This manuscript covers a lot of ground and makes a thorough analysis of the factors regulating the expression of these reporter genes. Specifically the role of cytokinins as well as sucrose and sugars in inducing *WUS* expression is shown and the obtained data suggest that the Target Of Rapamycin (TOR) kinase plays a central role in connecting metabolic signals to *WUS* induction. This manuscript is well written on the overall, although some Results parts could be shortened, and given the central and conserved roles of both TOR and *WUS/Clv3* in promoting organ formation in plants, it is likely to be of interest for a large audience.

Nevertheless, I have a few points that, to my opinion, should be addressed by the authors:

1) It is not clear from the Methods if the *WUS* 5'UTR is included in the chimaeric construct used, which should be more clearly defined? This could be important given the translation regulations of *WUS* expression (see Cui et al., PCE 2015). Thus I am not sure that a transcriptional fusion with a reporter gene completely recapitulates *WUS* or *Clv3* regulations.

2) Therefore I think that the data obtained with the fluorescent markers should be backed up, for the most important experiments, with another method like QPCR or even better immunolocalization. In situ hybridization was used but only for the first experiment.

3) The effect of TOR inhibition on *WUS* induction by sugars or light is quite clear and interesting. However, this effect could be indirect and linked to the pleiotropic effect of TOR inhibition on development and metabolism. This part of the paper is mainly descriptive and the authors could try to find more mechanistic evidence by mining the available data on the impact of TOR inhibition on transcriptome or metabolome. The western blot used to measure S6K phosphorylation, a classical readout for TOR activity, could be improved. The molecular mass of both phosphorylated and total S6K bands should be indicated together with a picture of the whole membrane. Much less proteins seem to have been loaded for dark-grown plants? Also a control experiment with AZD should be performed to check for lower S6K phosphorylation.

4) Some differences in metabolite abundance are quite small and should be confirmed by statistical analyses to avoid over-interpretation of the results.

---

## [Author Response]

All reviewers agree that your work is of general interest and in principle support publication. However, they also agree that a few issues have to be addressed before the work can be accepted. In your revision, please pay special attention to the following points:

1) The reviewers are worried about the validity of some of the pharmacological experiments because of possible indirect or side effects of the drugs (in particular of norfluorazon and lincomycin). Therefore, they believe that your paper would be substantially strengthened if you could repeat the basic introductory experiments (Figure 1 and Figure 2) in a CO_2_-free atmosphere.

We understand that the pharmacological experiments using the substances lincomyin and norflurazon bear the risk of triggering indirect side effects. We therefore now carried out additional experiments using CO_2_-deficient environment as proposed by reviewer #2 and described earlier in Kircher and Schopfer, 2012. The new experiments lead to the same conclusion as our previous experiments with norflurazon. The growth of plants in a CO_2_-deficient environment also inhibited development in light and could be rescued by the addition of sucrose to the medium (Figure 2—figure supplement 1). In 4 day old seedlings the CO_2_-deficiency did not affect light induced *WUS*-expression (Figure 2—figure supplement 1) but strongly suppressed the TCS reporter activity (Figure 3—figure supplement 1).

2) The reviewers also have some concerns regarding the plant material, especially with regard to the cop1 allele used. They ask that some of the basic experiments are verified with a cop1 null allele.

We used *cop1* mutants in several of our experiments. Since *cop1* null mutants arrest at early seedling stage (McNellis et al., 1994), we used the weak allele *cop1-4*. The *cop1-4* mutant expresses a truncated COP1 protein that lacks the WD repeats. A recent publication shows that the comparatively weak phenotype of the *cop1-4* mutant relies on the SPA proteins that can partially substitute the lack of the WD repeats that are missing in *cop1-4*_1_. In absence of the SPA proteins the COP1-4 protein has no activity.

Addressing the concerns of Reviewer #1, we want to point out that the development of *cop1* mutants in darkness is not restricted to the *cop1-4* allele. It has already been described in previous publications for the alleles *cop1-1, cop1-6* and *cop1-8* (Kircher & Schopfer, 2012; Nakagawa & Komeda, 2004).

Following the suggestions of the reviewers, we now repeated some key experiments with other *cop1* mutant alleles (namely *cop1*-6 (McNellis et al., 1994) and *cop1*-19 (Favory et al., 2009) and all confirmed the results obtained with *cop1-4*. We show that in all alleles tested *WUS* expression is induced in darkness (Figure 2—figure supplement 1) and that S6K phosphorylation levels are elevated (Figure 5—figure supplement 1). We therefore conclude that the effects are not restricted to the *cop1-4* allele, but characteristic for *cop1* mutants in general.

3) The reviewers acknowledge the use of reporters as an output for CLV3/WUS activity, but also wonder whether some of the basic observations could be confirmed by qPCR analysis of the endogenous genes?

Many conclusions of our manuscript are based on the analysis of the *WUS* reporter line because it is by far the most sensitive and robust way to quantify *WUS* promoter activity. We have characterized these reporters in great detail and found them to faithfully recapitulate in many different settings, including other transgenes, suggesting that positional effects have minor influence on their output (see Figure 6*pWUS:mCHERRY-GUS* reporter in 4 day old seedlings as an example).

Author response image 1.**DOI:**
http://dx.doi.org/10.7554/eLife.17023.015

As suggested by the reviewers, we have now quantified *WUS* expression by qRT-PCR to independently verify our results obtained with the reporter line. We were able to confirm our findings, however the lack of spatial resolution and the small number of expressing cells translated to low signal to noise ratio in the qRT assays. Therefore, we only detected mild expression changes of *WUS* with some variation in 4 day old seedlings grown in darkness, red light or in presence of sucrose (see Figure 7). We therefore decided to test 7 day old seedlings, which showed very reliable results (Figure 2—figure supplement 1 and E and Figure 4—figure supplement 1). The qRT-PCR analysis confirmed all key findings, including the induction of *WUS* expression by light and sucrose (Figure 2—figure supplement 1) as well as the positive effect of the *cop1* mutation (Figure 2—figure supplement 1) and the *ckx5/ckx6* double mutant (Figure 4—figure supplement 1) on *WUS* expression.

We want to point out that quantitative analysis of *WUS* expression by qRT-PCR is problematic, since we need to compare plants of different morphology and thus an appropriate reference gene is not available (the issue was discussed in detail by Demidenko and Penin, 2012 _2_). We believe it to be essential to analyze WUS expression with cellular resolution and thus view the qRT-PCR results as a qualitative rather than a quantitative confirmation of our reporter gene analysis.

Author response image 2.**DOI:**
http://dx.doi.org/10.7554/eLife.17023.016

Please find the reviewer comments attached below for more details, and please respond to them to the best of your capacity.

Reviewer #1:

This manuscript from the Lohmann lab and colleagues presents a complex set of data, which indicate that SAM activation is both light- and energy-status dependent in Arabidopsis. Overall, the data appear solid and are presented in a transparent manner, and the conclusions are largely justified from my point of view. However, the authors' conclusions strongly hinge on the quantification of two reporters under different conditions or in different genetic backgrounds, and I believe some improvements are required to make the manuscript acceptable for publication.

1) I have one major concern with respect to the genetic material. Several analyses are based on experiments with a mutant in COP1, a central negative regulator of photomorphogenesis. Understandably, the authors use one of the viable cop1 alleles in their work, the cop1-4 mutant. However, within the cop1 allelic series, this allele has always been the most mysterious. Unlike true null alleles, cop1-4 produces a truncated protein that comprises both the RING finger and coiled coil domains, yet alleles that retain these domains as well but are truncated further C-terminal or carry certain point mutations in the WD40 repeat domain also have a strong phenotype.

Therefore, I believe it is important to check whether the results obtained with cop1-4 are representative. I understand that it would be too much to ask to repeat reporter experiments in another cop1 background, but I think the authors could check the principal physiological responses in true cop1 null alleles, which should be easily available. In particular, the authors could repeat the experiment of Figure 1 and Figure 5. They could also try to detect the effect of cop1 on WUS expression by qPCR instead of a reporter, the difference appears strong enough although it is restricted to few cells (Figure 2).

Done – Please see response to Reviewing Editor point 2 above.

2) Another genetic concern: I believe the authors are formally right to conclude that HY5 has no role in light-dependent stem cell activation, however it is important to remember that unlike cop1, none of the COP1-target mutants has a phenotype that is as absolute skotomorphogenic as cop1 is photomorphogenic. If their conclusions with respect to cop1 are right, multiple mutants combining various COP1 targets should eventually show the pertinent defect. Among others, the hy5 hyh double mutant comes to mind. In the shoot, HYH is stronger expressed than HY5, and hy5 hyh double mutants have been reported to have meristematic defects, like "delayed" organ formation. I believe the authors should check light-dependent SAM activation at least in hyh single and hy5 hyh double mutants, and maybe also pertinent triple mutants with additional factors.

The hy5 mutation did not have any effect WUS expression in our reporter line. However, Reviewer #1 is right in pointing out that other COP1 targets might be also involved in downstream signaling and therefore the effects of the hy5 mutation might be masked by redundancy. Since we cannot exclude that HY5 is involved in regulation of meristem activity, we now state in the text:

"However, hy5 mutants showed no defect in activation of WUS expression (Figure 2—figure supplement 1), suggesting that either SAM stem cell activation makes use of another COP1-targeted transcriptional transducer or the function of HY5 can in this case be substituted by the HY5 HOMOLOG (HYH)."

and in the Discussion:

"However, we cannot exclude that the HY5 homolog HYH with partially overlapping function but higher expression level in the shoot might mask any effect of *hy5* on *WUS* expression (Holm et al., 2002; Sibout et al., 2006). Since *hy5/hyh* mutants showed deficiencies in development of the first leaved future experiments analyzing such a double mutant might give further insights."

We currently prepare CRISPR-CAS9 mutants of HYH in the *hy5* double reporter background to test this hypothesis. However, these experiments will take six months to carefully execute and go well beyond the scope of this manuscript.

3) With respect to responses illustrated by images only, like cop1 shoot formation on +/- suc, are those responses fully penetrant? If so, this should be explicitly stated, if not, proportions clearly indicated.

We now clarify the percentage of plants with shoot formation in the text. We see that 100% of *cop1-4* mutants developing leaves on sucrose. In contrast, only approximately one third of WT plants develop leaves on norflurazon CO_2_-deficient environment in light when supplemented with sucrose. In the absence of sucrose we never see development of the *cop1* mutants in darkness or WT on norflurazon or CO_2_-deficient environment in light.

4) One of the two ML1 transgenic lines deviates from the highlighted phenotype (Figure 2—figure supplement 1). Why is that? Is it an expression problem? As I understand, the phyB mutant was not tagged? Maybe the authors can present evidence from additional lines and/or correlation with expression levels. As is, their statement regarding ML1 is misleading, because one out of two lines shows the other phenotype, so it's a 50:50 choice.

When we grew *pML1:PHYB Y276H* expressing lines in darkness, some showed a *cop*-like phenotype, while others resembled WT seedlings. We now show that this correlates with the expression level of *PHYB* by analyzing the total and endogenous *PHYB* expression by qRT PCR. Line AP88.2.1 (*cop* phenotype in darkness) shows 15x higher level of *PHYB* than line AP88.1.3 (WT phenotype) (Figure 2—figure supplement 1). Additionally we also analyzed the expression level of several lines expressing *pML1:PHYB Y276H-YFP* and also saw a clear negative correlation of hypocotyl length and *PHYB Y276-YFP* expression strength (see Figure 8 pictures of 2 seedlings of each line grown for 4 days in darkness and above each line corresponding epifluorescence pictures of the YFP signal of hypocotyl cells with intensity coded in FIJI LUT "fire").

Author response image 3.**DOI:**
http://dx.doi.org/10.7554/eLife.17023.017

5) With regard to the cytokinin experiments, and the sometimes inconclusive contribution, I wonder whether the actual cytokinin used is of importance? Given the variety of cytokinins and their reported shoot- and root-specific activities, the authors might find a "better" cytokinin, which might have a more clear effect on WUS expression.

To elucidate the role of different cytokinin molecules we now assayed the effect of Kinetin and Zeatin on *WUS* expression. However, we did not see striking differences in the activation of *WUS* expression (see Figure 9, plants were grown for 6 days in darkness and treated with 0.5 µM of cytokinin during the last 3 days).

Author response image 4.**DOI:**
http://dx.doi.org/10.7554/eLife.17023.018

6) Because of the importance of the fluorescence measurements, I would like the authors to confirm that samples analyzed for one graph were grown in parallel and analyzed with identical settings.

All datasets were grown and analyzed in parallel with the reference sample (which is either the red light treated sample or the glucose treated sample) as well as an untreated (dark grown) sample and we now state so explicitly in the Materials and methods section:

"For all these experiments plants of one experimental set were always grown and analyzed in parallel to the untreated (dark-grown) and the corresponding reference sample."

We already mentioned in the Methods that microscope settings were identical during all measurements. To illustrate that the measurements were highly reproducible we present an example of two experiments, which were *not* done at the same time. The plot shows *WUS* expression in dark-grown WT, red-light-grown WT and red-light-grown *phyB* mutants grown and analyzed days apart (box 1-3 and box 4-6).

Author response image 5.**DOI:**
http://dx.doi.org/10.7554/eLife.17023.019

Reviewer #2:

1) Based on WUS expression in cop1 +/- sucrose and WT +/- light and the use of NF and Linc (Figure 2), the authors conclude that both light and metabolic signals contribute to WUS expression. How can we be sure that NF and Linc were really effective in the experiment shown in 2A? More generally, if at all possible it would more elegant/convincing to show the role of light on the production of sugars by performing at least one experiment in a CO_2_ free environment (see for example Kircher and Schopfer 2012). This would not lead to the rather drastic effects of these drugs on chloroplasts. In principle the effect of CO_2_-free air could be complemented by adding sucrose. Haydon et al., 2013 also complemented the NF effect by adding sucrose which to some extent rules out non-specific effects of NF.

We want to thank Reviewer #2 for pointing out that growth a CO_2_-free environment would be a good alternative for the norflurazon and lincomycin treatments. We now have carried out the suggested experiments and found them to confirm our earlier results. For further details please see response to Reviewing Editor point 1 above.

2) The role of CK is potentially quite complex and I'm not sure that the current data allows drawing a clear conclusion. The authors propose that CK promote WUS expression by transmitting the light signal (e.g. subsection “Mechanisms of hormonal stem cell activation”, end of last paragraph). In Figure 2 one can observe that light-grown plants treated with NF still show strong WUS expression. However, on Figure 3 light grown seedlings treated with NF show absolutely no TCS:GUS signal, indicating no CK activity. How then could such seedlings express WUS (Figure 2) given that the metabolic signal should be gone (in the presence of NF) and TCS:GUS is also gone? A second problem with the model is the role of light-regulated CKX5/6 expression. The authors propose that this is how light regulates the CK function in this process. However, light regulation of WUS expression is just as robust in the ckx5/ckx6 double mutant than in the WT (4B). It therefore rather appears that these genes are involved in basal regulation of WUS expression and not in light regulation. It would actually be interesting to test this by adding sucrose to the double mutant and look for signs of leaf development in darkness rather than hypocotyl elongation as shown in supplementary Figure 4.

We appreciate the reviewers' thought on this complex matter and agree that the proposed role of CK in transmitting light signals likely is only one of many.

We have carried out the experiment suggested to test a potential role of CK as a regulator of basal WUS expression. However, *ckx5/ckx6* mutants grown in darkness for a prolonged time do not develop leaves, even in the presence of sucrose (see Figure 4—figure supplement 1). Therefore, additional development promoting components are still missing in line with the CKX genes being only one element in the system. The reviewer is also right to point out the discrepancy between WUS activation and TCS activity, but since TCS is merely a read out B-type ARR activity and not of CK levels, and we have no knowledge on the relative sensitivities of the two promoters, or the directness of WUS regulation, we are unable to provide any further mechanistic insight into that matter at the moment.

We pointed out these complications in the following paragraph of the Discussion: "Also the activity of the CK output reporter *pTCSn:GUS* strictly depended on sucrose, but could be further stimulated by light, suggesting that CK signaling mainly acts to transmit light signals, but that elevation of CK levels genetically or by treatment are insufficient to elicit the full developmental response. Based on these results we suggest that light enhances CK levels by reducing the expression of CKX genes, however we cannot rule out other explanations, such as stimulation of CK biosynthesis."

3) A potential difficulty with the role of CK in this process and the use of drugs like NF is that CK biosynthesis requires plastids and chloroplast development is promoted by CK. Not sure how this can be untangled but it should be discussed.

To disentangle the dependency of CK biosynthesis on chloroplast development we now grew plant in a CO_2_-free atmosphere in addition to the pharmacological treatment to show the dependency of CK signaling on energy and this new experiment confirmed our original findings (Figure 3—figure supplement 1).

4) All figure legends should be more explicit and provide detailed information about the exact light conditions (intensity, color), presence of sucrose and numbers of seedlings used per experiment. For example, I don't see which data correspond to + sucrose (hatched box) in 2A.

We tried to improve the readability of the figures and now add more information about the growth conditions in the figure legends. We also refer to the Materials and methods section, where we explained in detail the quantification of the fluorescence intensities.

Reviewer #3:

This interesting manuscript investigates the regulation of the stem cell inducer WUSCHEL (WUS) and of it repressor CLAVATA3 (Clv3) by metabolic, hormonal and light signals using fluorescent markers driven by the promoters of these genes. This manuscript covers a lot of ground and makes a thorough analysis of the factors regulating the expression of these reporter genes. Specifically the role of cytokinins as well as sucrose and sugars in inducing WUS expression is shown and the obtained data suggest that the Target Of Rapamycin (TOR) kinase plays a central role in connecting metabolic signals to WUS induction. This manuscript is well written on the overall, although some Results parts could be shortened, and given the central and conserved roles of both TOR and WUS/Clv3 in promoting organ formation in plants, it is likely to be of interest for a large audience.

Nevertheless, I have a few points that, to my opinion, should be addressed by the authors:

1) It is not clear from the Methods if the WUS 5'UTR is included in the chimaeric construct used, which should be more clearly defined? This could be important given the translation regulations of WUS expression (see Cui et al., PCE 2015). Thus I am not sure that a transcriptional fusion with a reporter gene completely recapitulates WUS or Clv3 regulations.

Reviewer #3 pointed out the importance of the 5'UTR in the regulation of WUS expression. The 5' and 3´ UTRs are part of our *WUS* reporter and the regulatory regions used in the reporter have been shown to fully rescue the meristem defect of *wus* mutants in the context of a WUS-GFP fusion, which has been published and is also used in our study (*pWUS:WUS- linker-GFP* in *wus* mutant background (Daum et al., 2014). To clarify the reporter design, we now state in the text:

"The *pWUS:3xVENUS-NLS* construct (pTS81) was generated accordingly, using a pGREEN- IIS based binary vector with a 4.4 kb genomic *WUS* promoter upstream of the ATG and 2.8 kb *WUS* terminator downstream of the stop codon (pFK398) and a Basta resistance cassette."

2) Therefore I think that the data obtained with the fluorescent markers should be backed up, for the most important experiments, with another method like QPCR or even better immunolocalization. In situ hybridization was used but only for the first experiment.

We have now repeated key experiments using qRT-PCR and confirmed our results obtained with the WUS reporter. For a detailed answer, please see response to Reviewing Editor point 3 above.

3) The effect of TOR inhibition on WUS induction by sugars or light is quite clear and interesting. However, this effect could be indirect and linked to the pleiotropic effect of TOR inhibition on development and metabolism. This part of the paper is mainly descriptive and the authors could try to find more mechanistic evidence by mining the available data on the impact of TOR inhibition on transcriptome or metabolome. The western blot used to measure S6K phosphorylation, a classical readout for TOR activity, could be improved. The molecular mass of both phosphorylated and total S6K bands should be indicated together with a picture of the whole membrane. Much less proteins seem to have been loaded for dark-grown plants? Also a control experiment with AZD should be performed to check for lower S6K phosphorylation.

Normalization of TOR activation assays is challenging in situations of diverging protein synthesis, such as light vs. dark, since TOR kinase has been described as a central regulator of protein synthesis. Therefore, we decided to load equal amounts of fresh weight to the gel instead of equal total protein amounts and to use unphosphorylated S6K1/2 for normalization. In our experience loading equal fresh weight results in rather similar amounts of unphosphorylated S6K1/2 signal, while protein levels correlated strongly with TOR activity, as expected. We now repeated the Western Blot analysis on AZD-8055 treated samples as control to confirm the specificity of the phosphorylation specific antibody (Figure 5—figure supplement 1). As requested, we now show a larger section of the membrane and added indicators of molecular mass.

As suggested, we have again screened the literature and databases for candidates that might explain the effect of TOR on SAM development. Obviously, several regulators could be identified, but since we cannot provide any evidence of the contribution of these candidates in meristem regulation we would like to constrain ourselves from speculation at this point.

We hope that the manuscript text makes this point rather clear:

"Characteristic expression changes that were described in response to glucose-TOR signaling (Xiong et al., 2013) and E2Fa overexpression (Vandepoele et al., 2005; López- Juez et al., 2008), namely affecting genes involved in ribosome biogenesis, protein translation and cell proliferation have also been identified in microarray analyses of shoot apex tissue derived from young seedlings (López-Juez et al., 2008). These genes were rapidly and synchronously induced by photosynthetically active light preceding organ growth, which lead us to hypothesize that stem cell activation in the SAM might also be under control of the TOR kinase."

4) Some differences in metabolite abundance are quite small and should be confirmed by statistical analyses to avoid over-interpretation of the results.

We now added statistical information to all metabolite measurements and also added a PCA that analyses the overall variability of our data sets and replicates.

References

1) Ordoñez-Herrera, N. et al. A cop1 spa Mutant Deficient in COP1 and SPA Proteins Reveals Partial Co-Action of COP1 and SPA during *Arabidopsis* Post-Embryonic Development and Photomorphogenesis. Mol. Plant 8, 479–481 (2015).

2) Demidenko, N. V. & Penin, A. A. Comparative analysis of gene expression level by quantitative real-time PCR has limited application in objects with different morphology. PLoS One 7, 5–10 (2012).